# Perturbation of protein homeostasis brings plastids at the crossroad between repair and dismantling

**Luca Tadini[1]☉, Nicolaj Jeran[1]☉, Guido Domingo[2], Federico Zambelli ⬤[1], Simona Masiero[1], Anna Calabritto[1¤a], Elena Costantini[2], Sara Forlani[1¤b], Milena Marsoni[2], Federica Briani[1], Candida Vannini[2], Paolo Pesaresi[1] ***

**1** Dipartimento di Bioscienze, Università degli Studi di Milano, Milano, Italy, **2** Dipartimento di Biotecnologie e Scienze della Vita, Università degli Studi dell'Insubria, Varese, Italy

☉ These authors contributed equally to this work.
¤a Current address: Department of Physics and Astronomy, Vrije Universiteit, Amsterdam, Amsterdam, the Netherlands
¤b Current address: Faculty of Biological and Environmental Sciences, University of Helsinki, Helsinki, Finland
* paolo.pesaresi@unimi.it

**Data Availability Statement:** The data that support the findings of this study are openly available in Gene Expression Omnibus at https://www.ncbi.nlm.nih.gov/geo/, reference number GSE205271, and in ProteomeXchange Consortium at http://

## Abstract

The chloroplast proteome is a dynamic mosaic of plastid- and nuclear-encoded proteins. Plastid protein homeostasis is maintained through the balance between *de novo* synthesis and proteolysis. Intracellular communication pathways, including the plastid-to-nucleus signalling and the protein homeostasis machinery, made of stromal chaperones and proteases, shape chloroplast proteome based on developmental and physiological needs. However, the maintenance of fully functional chloroplasts is costly and under specific stress conditions the degradation of damaged chloroplasts is essential to the maintenance of a healthy population of photosynthesising organelles while promoting nutrient redistribution to sink tissues. In this work, we have addressed this complex regulatory chloroplast-quality-control pathway by modulating the expression of two nuclear genes encoding plastid ribosomal proteins PRPS1 and PRPL4. By transcriptomics, proteomics and transmission electron microscopy analyses, we show that the increased expression of *PRPS1* gene leads to chloroplast degradation and early flowering, as an escape strategy from stress. On the contrary, the overaccumulation of PRPL4 protein is kept under control by increasing the amount of plastid chaperones and components of the unfolded protein response (cpUPR) regulatory mechanism. This study advances our understanding of molecular mechanisms underlying chloroplast retrograde communication and provides new insights into cellular responses to impaired plastid protein homeostasis.

## Author summary

Chloroplast protein composition requires the contribution of both nuclear and plastid genomes. The homeostasis of the plastid proteome is preserved through the balance

www.proteomexchange.org/, reference number PXD034479.

**Funding:** This research was funded by MUR—Ministero dell'Università e della Ricerca, grant number PRIN 2017-FBS8YN, entitled to PP. The funders had no role in study design, data collection and analysis, decision to publish, or preparation of the manuscript.

**Competing interests:** The authors have declared that no competing interests exist.

between de novo protein synthesis and protein degradation, while plastid-to-nucleus communication mechanisms ensure the correct coordination of the two genomes. Chloroplast maintenance is costly; therefore, the degradation of damaged chloroplasts allows nutrient redistribution, which contributes to the essential requirement to sustain a population of healthy chloroplasts able to perform photosynthesis. In this work, we studied chloroplast quality-control mechanisms by modulating the expression of two nuclear genes encoding the plastid ribosomal proteins PRPS1 and PRPL4. By a multidisciplinary approach, we show that the increased expression of *PRPS1* leads to chloroplast degradation and early flowering, as a stress-escaping strategy. On the contrary, the over-accumulation of PRPL4 protein activates the unfolded protein response mechanisms. Overall, this study provides further hints toward the comprehension of the molecular mechanisms underlying chloroplast retrograde communication and plastid protein homeostasis.

## Introduction

Chloroplasts are plant cell organelles of cyanobacterial origin that perform essential metabolic and biosynthetic functions including photosynthesis and fatty acid biosynthesis. The Arabidopsis chloroplast proteome is estimated to consist of several thousand proteins, most of which are encoded by the nuclear genome and post-translationally imported into the organelle [1]. The plastid genome, encoding about a hundred proteins, is expressed by transcriptional and translational machineries that conserve many bacteria-like elements. For instance, the chloroplast ribosome shares several features with that of the model organism *E. coli*, used as the reference in early investigations [2]. Almost all the chloroplast ribosomal proteins have an orthologue in *E. coli*, however, few differences in ribosome composition, protein domain organisation and function have been described [3–10].

This is the case, for instance, of the S1 protein, encoded by *rpsA* gene, the largest ribosomal protein present in *E. coli* 30S subunit essential for cell viability [11]. S1 promotes the translation initiation step by recognising diverse mRNA leaders and mediating their interaction with the ribosome [12,13]. Furthermore, S1 is found in *E. coli* cells both as ribosome-associated- and as free-subunit in cytoplasm [14–16], and it is responsible for its own post-transcriptional regulation [17]. S1 bears six homologous non-identical repeats known as S1 domains, members of an ancient RNA binding OB-fold family [14,18]. It has been proposed that the six S1 domains have a partial functional specialisation, which correlates with their relative position [19]. In particular, domains 1 and 2 mediate the interaction with the ribosome [20,21], domains 3, 4, 5 and 6 are responsible for RNA binding and unfolding [14,22,23], with domains 5 and 6 also involved in transcription stimulation and autoregulation [24,25]. Recently, domains 4 and 6 have been shown to be implicated in ribosome dimerization and hibernation under stress [26].

Unlike S1, the *Arabidopsis thaliana* Plastid Ribosomal Protein Small subunit 1 (PRPS1) is characterised by three S1 domains. Its function was characterised by exploiting the knock-down allele *prps1-1*, caused by a T-DNA insertion +3 bp downstream the transcription starting site, which produces about one-tenth of wild-type PRPS1 transcripts resulting in one-third of wild-type PRPS1 protein levels in adult plants. Such impairment in *prps1-1* mutants affects the overall plant growth rate and results in pale leaves due to decreased translation in chloroplasts [27]. *A. thaliana* PRPS1 has been found to genetically and physically interact with the nuclear-encoded plastid protein GUN1 [28]. As a consequence, the depletion of GUN1 in the *prps1-1* genetic background leads to the partial rescue of the mutant phenotype, restoring to wild-

type-like levels both PRPS1 abundance and the chloroplast translation capacity [28]. These results indicate a direct negative regulation of PRPS1, and therefore of chloroplast translation, by GUN1, possibly due to its relations with the chloroplast protein homeostasis machinery [28,29]. Further investigations revealed the involvement of PRPS1 and plastid translation in retrograde signalling upon heat-stress. Under such conditions, the diminished chloroplast translational capacity of *prps1-1* prevents the up-regulation of the *HSFA2* gene, a master regulator of the chloroplast Unfolded Protein Response [30,31]. Accordingly, seedlings and adult *prps1-1* plants are unable to cope with high temperature showing low survival rates with respect to the wild-type [31]. Interestingly, the introgression of *gun1* mutation in *prps1-1* genetic background rescues its low survival rate in such conditions [28]. In addition, attempts to constitutively overexpress *PRPS1* gene resulted in a virescent phenotype and in the decrement of PRPS1 protein levels [31].

Taken together, these observations indicate that chloroplast gene expression is largely controlled at the translational and post-translational level and set the chloroplast translation as a crucial step for the genesis of chloroplast-to-nucleus retrograde signalling pathways [8,32]. Furthermore, the peculiar features of ribosomal protein S1 identified during studies on *E. coli* and *Arabidopsis thaliana*, make it an interesting subject for deeper investigations regarding its role in chloroplast biogenesis, translational regulation and the interconnection with the protein homeostasis maintenance and retrograde communication.

These aspects have been investigated in the present manuscript, where we demonstrate that the knock-out of *PRPS1* gene is incompatible with chloroplast biogenesis and embryo development. Further, we assessed that PRPS1 is unable to functionally replace S1 in *E. coli* cells, and its overexpression inhibits cell growth, as in the case of the endogenous S1 protein [16,33]. We also show that PRPS1 protein accumulation in chloroplasts is controlled post-translationally by the plastid CLP protease complex, while its constitutive over-expression promotes chloroplast degradation via micro- and macro-autophagy [34], and induces early flowering. This adaptive response is organised at the very beginning of *PRPS1* transcript over-accumulation, as revealed by the transcriptome profile of short-term induced *PRPS1* expression lines. On the contrary, the over-accumulation of Plastid Ribosomal Protein Large subunit 4 (PRPL4) [27,35], here used as control, is tolerated by chloroplasts and leads to the accumulation of transcripts and proteins, such as chaperons and proteases, that are part of the chloroplast-derived Unfolded Protein Response mechanism (cpUPR) [36–39].

## Materials and methods

### Bioinformatic analyses

S1 domain sequences have been identified using InterPro online tool (https://www.ebi.ac.uk/interpro/). Multiple sequence alignment was performed with MUSCLE online tool (https://www.ebi.ac.uk/Tools/msa/muscle/) and represented as phylogenetic tree employing PhyML (https://toolkit.tuebingen.mpg.de/tools/phyml) and iTOL (https://itol.embl.de/).

### Plant material and growth conditions

The *PRPS1/prps1-2* heterozygous mutant lines were generated by targeting the first exon of *PRPS1* locus in *Arabidopsis thaliana* wild-type (Col-0) genetic background, using the pDe-CAS9 vector earlier described (guide RNA sequence is listed in S1 Table) [40]. *PRPS1/prps1-2* heterozygous plants, devoid of CAS9 T-DNA, were selected based on the mutation in *PRPS1* sequence. *prps1-2 pPRPS1::PRPS1* complemented lines were obtained by introgressing *PRPS1* genomic locus, including the *pPRPS1* promoter region, in *PRPS1/prps1-2* heterozygous plants and by selecting *prps1-2* viable plants carrying the *pPRPS1::PRPS1* construct. *oePRPS1* and

*oePRPL4* lines were obtained by Agrobacterium-mediated transformation of Arabidopsis Col-0 genetic background with the coding sequences of *PRPS1* and *PRPL4* genes, under the control of CaMV35S promoter (pB2GW7 plasmid; https://gatewayvectors.vib.be/). *indPRPS1* and *indPRPL4* lines were obtained by cloning the *PRPS1* and *PRPL4* coding sequences in the Dexamethasone (DEX)-inducible *pOp/LhG4* system [41] and by Agrobacterium transformation of Arabidopsis Col-0 plants. *prps1-1* (SAIL_560_B02), *prpl11-1* (GABI_380H05), *clpc1-1* (SALK_014058C) and *clpd-1* (SALK_110649C) T-DNA lines were described in previous works [27,42–44] and manually crossed for obtaining *prps1-1 prpl11-1*, *prps1-1 clpc1-1*, *prps1-1 clpd-1* double mutants. Primers required for gene cloning and mutant line isolation are listed in S1 Table. Wild-type and mutant seeds were grown on soil in climate chambers under long-day (150 µmol m$^{-2}$ s$^{-1}$ 16 h/8 h light/dark cycles) and short-day (150 µmol m$^{-2}$ s$^{-1}$ 8 h/16 h light/dark cycles) conditions. For growth experiments on Dexamethasone, seeds were surface-sterilised and grown for 16 days (80 µmol m$^{-2}$ s$^{-1}$ on a 16 h/8 h light/dark cycle) on Murashige and Skoog medium (Duchefa) supplemented with 1% (w/v) sucrose and 1.5% Phyto-Agar (Duchefa), Dexamethasone was added at the final concentration of 2 µM. Growth rate was determined by ImageJ software (https://imagej.nih.gov/).

## Whole-mount preparation and optical microscopy

To analyse defects in embryo development, siliques of Col-0 and heterozygous *PRPS1/prps1-2* plants were manually dissected and cleared as reported previously [45]. Developing seeds were observed using a Zeiss Axiophot D1 microscope equipped with differential interface contrast optics. Images were documented with an Axiocam MRc5 camera (Zeiss).

## *E. coli* strains

*PRPS1* coding sequence was cloned into pQE31-pREP4 plasmid system (primers are listed in S1 Table), under the control of bacteriophage T5 promoter fused upstream to *lacO* operator sequences. *PRPS1-pQE31-pREP4* and *rpsA-pQE31-pREP4* plasmids [33] were then transferred into *araBp-rpsA* conditional expression strain C-5699, in which the *rpsA* gene is expressed in presence of 1% (w/v) arabinose and repressed in presence of 0.4% (w/v) glucose. To obtain the *PRPS1* overexpressing strain, *PRPS1-pQE31-pREP4* plasmids were introduced into the *E. coli* C-1a strain [46], while *rpsA-pQE31-pREP4* strain C-5691 was used as control [16,33].

## Chlorophyll a fluorescence measurements

The Imaging Chl a fluorometer (Walz Imaging PAM; https://www.walz.com/) was used to determine Chl fluorescence *in vivo*. Eight plants for each genotype and condition were analysed at 18 days after sowing (DAS). Average values plus-minus standard deviations were then calculated. 20 min dark-adapted plants were exposed to blue measuring light (intensity 4) and a saturating light flash (intensity 4) was used to calculate the maximum quantum yield of PSII, *Fv/Fm*.

## Nucleic acid analyses

For qRT-PCR analyses, 1 µg of total RNA was treated with iScript™ gDNA Clear cDNA Synthesis Kit (Bio-Rad; https://www.bio-rad.com/) for genomic DNA digestion and first-strand cDNA synthesis. qRT-PCR analyses were performed on a CFX96 Real-Time system (Bio-Rad; https://www.bio-rad.com/), using primer pairs listed in S1 Table. *PP2AA3* (*AT1G13320*) transcripts were used as internal reference [47]. Data obtained from three biological and three

technical replicates for each sample were analysed with the Bio-Rad CFX Maestro 1.1 (v 4.1) (Bio-Rad; https://www.bio-rad.com/).

### *In vivo* Translation Assay

The *in vivo* translation assay was performed essentially as previously described [48]. *indPRPS1* leaf discs (6 mm in diameter) were vacuum-infiltrated in liquid MS medium supplemented with 1% (w/v) sucrose and, where indicated, 2 μM Dexamethasone. After 6-hours exposure to 80 μmol photons $m^{-2} s^{-1}$ white light, leaf discs were incubated with a buffer containing 1 mM $K_2HPO_4$–$KH_2PO_4$ (pH 6.3) and 0.05% (v/v) Tween-20, supplemented with 20 μg/ml cyclohex-imide, to inhibit cytosolic translation. [$^{35}$S]methionine was then added (0.1 mCi/ml) and leaf discs were vacuum-infiltrated and exposed to light (80 μmol photons $m^{-2} s^{-1}$). 5 leaf discs were collected at each time point (15 and 30 min). Total protein extraction and Tris-glycine SDS-PAGE fractionation is described below. Signals were detected using the Phosphorimager GE Healthcare Life Sciences (https://www.gehealthcare.com/).

### Isolation of PRPS1-containing protein complexes

The isolation of PRPS1-containing complexes was performed according to previous works [49,50]. 100 mg of leaf fresh weight were ground in liquid nitrogen and resuspended in 1 ml 0.2 M Tris-HCl, pH 9, 0.2 M KC1, 35 mM $MgC1_2$, 25 mM EGTA, 0.2 M sucrose, 1% Triton X-100, 2% polyoxyethylene-10-tridecyl ether, supplemented with 500 μg/ml heparin, 100 μg/ml chloramphenicol and 25 μg/ml cycloheximide. The extract was then solubilised with 0.5% (w/v) sodium deoxycholate for 5 min on ice. After centrifugation (15 min at 10000g), 800 μl of supernatant was loaded onto 3.6 ml 15–55% (w/v) sucrose gradients in polysome gradient buffer (40 mM TrisHCl pH 8, 20 mM KCl, 10 mM $MgCl_2$, 100 μg/ml chloramphenicol and 500 μg/ml heparin). Sucrose gradients were centrifuged in SW60 rotors (Beckman) for 18 h at 180000 g at 4˚C. 9 fractions of 400 μl each were collected from the top of the tube and subjected to SDS-PAGE fractionation. *E. coli* ribosome fractionation was performed as described [16]. Quantification of PRPS1 protein accumulation in Low Molecular Weight (LMW; fractions 1–5) and High Molecular Weight (HMW, fractions 6–9) fractions was evaluated by using the Image Lab software on representative blots obtained from three biological and nine technical replicates. Each lane has been quantified as absolute value. The sum of all signals was set to 1. The values reported in the graph are relative to the total sum for each condition (DEX -/+).

### Transmission electron microscopy (TEM)

TEM analyses were performed as described previously [51]. Plants were grown for 16 days on MS synthetic medium supplemented with 1% (w/v) sucrose and, where indicated, 2 μM Dexa-methasone. Plant material was vacuum-infiltrated with 2.5% glutaraldehyde, in 100 mM sodium cacodylate buffer, for 4 h at room temperature and incubated overnight at 4˚C. Samples were rinsed twice with 100 mM sodium cacodylate buffer for 10 min each, and post-fixed in 1% (w/v) osmium tetroxide in 100 mM cacodylate buffer for 2 h at 4˚C. After washings, samples were counterstained with 0.5% (w/v) uranyl acetate overnight at 4˚C, in the dark. The tissues were then dehydrated by increasing concentrations of ethanol (70%, 80%, 90%; v/v), 10 min each. Samples were then dehydrated with 100% ethanol for 15 min and permeated twice with 100% propylene oxide for 15 min. Epon-Araldite resin was prepared mixing properly Embed-812, Araldite 502, dodecenylsuccinic anhydride (DDSA) and Epon Accelerator DMP-30. Samples were infiltrated first with a 1:2 mixture of Epon-Araldite and propylene oxide for 2 h, then with Epon-Araldite and propylene oxide (1:1) for 1 h and left in a 2:1 mixture of Epon-Araldite and propylene oxide overnight at room temperature. Samples were then

incubated in pure resin before polymerisation at 60˚C for 48 h. Ultra-thin sections of 70 nm were then cut with a diamond knife (Ultra 45˚, DIATOME) and collected on copper grids (G300-Cu, Electron Microscopy Sciences). Samples were observed by transmission electron microscopy (Talos L120C, Thermo Fisher Scientific) at 120 kV. Images were acquired with a digital camera (Ceta CMOS Camera, Thermo Fisher Scientific).

## Protein sample preparation and immunoblot analyses

For immunoblot analyses, total proteins were prepared as described [52]. Plant material was homogenised in Laemmli sample buffer [20% (v/v) glycerol, 4% (w/v) SDS, 160 mm Tris–HCl pH 6.8, 10% (v/v) 2-mercaptoethanol] to a concentration of 0.1 mg μl$^{-1}$ (leaf fresh weight/ Laemmli sample buffer). Samples were incubated for 15 min at 65˚C and, after a centrifugation step (10 min at 16 000 g), the supernatant was incubated for 5 min at 95˚C. Protein samples corresponding to 4 mg (fresh weight) of seedlings were fractionated by SDS–PAGE 10% (w/v) acrylamide [53] and then transferred to polyvinylidene difluoride (PVDF) membranes (0.45 μm pore size). Replicate filters were immunodecorated with specific antibodies. Antibodies directed against AtHsp90-1 (AS08 346) and ClpB3 (AS09 459) were obtained from Agrisera (https://www.agrisera.com/), AtHsc70-4 antibody was obtained from Antibodies-online (https://www.antibodies-online.com/), antibodies directed against plastid ribosomal proteins (PRPS1, PRPL4 and PRPS5) were obtained from Uniplastomic, while polyclonal S1 antibody was kindly donated by U. Bläsi (University of Vienna).

## Transcriptome analysis

Total RNA was extracted from leaf discs harvested from *indPRPS1* and *indPRPL4* plants and vacuum infiltrated in either the absence or presence of DEX for 6 hours for a total of 5 biological replicates for each group. Total RNA extraction was performed using RNeasy Mini Kit (Qiagen), according to the manufacturer's instructions. RNA concentrations and integrity were determined via NanoDrop One C (ThermoFisher Scientific) and agarose-gel electrophoresis. Extracted RNA samples were sent to Novogene for sequencing via high throughput Illumina NovaSeq platform which employ a paired-end 150 bp sequencing strategy. The RNA-seq library was prepared by enriching for the PolyA-containing transcripts, therefore, no plastid-encoded genes are represented in the library. Raw data were processed and mapped to the Arabidopsis genome TAIR10 using STAR-RSEM software [54]. Differentially Expressed Genes (DEGs) were identified through the R package EdgeR (v 3.15) [55]. Called DEGs were statistically filtered via Benjamini-Hochberg False Discovery Rate method (FDR < 0.05). GO enrichment analyses were performed using agriGO v2.0 online tool and further processed by REVIGO [56,57]. The RNA-seq data were deposited in the Gene Expression Omnibus data repository under the dataset identifier GSE205271.

## Proteome analysis

Proteins were extracted from 1 g of plantlets following SDS/phenol method as described [58]. Proteins were then digested with trypsin via Filter Aided Sample Preparation (FASP) [59]. Peptides were analysed by LC-MS/MS as described [60]. Briefly, after LC separation peptides were sprayed into the mass spectrometer and eluting ions were measured in an Orbitrap mass analyser set at a resolution of 35000 and scanned between m/z 380 and 1500. Data dependent scans (top 20) were employed to automatically isolate and generate fragment ions by higher energy collisional dissociation (HCD); Normalised collision energy (NCE): 25% in the HCD collision cell and measurement of the resulting fragment ions was performed in the Orbitrap analyser, set at a resolution of 17500. Peptide ions with charge states of 2$^+$ and above were

selected for fragmentation. Raw data were searched against the *Arabidopsis thaliana* TAIR protein database (2010 version) with MaxQuant program (v.1.5.3.3), using default parameters. For the quantitative analysis, the "ProteinGroups" output files were filtered to retain only protein groups detected with at least two peptides in at least three of the four biological replicates, and in at least one analytical group. The mass spectrometry proteomics data have been deposited in the ProteomeXchange Consortium via the PRIDE [61] partner repository under the dataset identifier PXD034479. Missing values were replaced with the R package imputeLCMD (v.2.1) using Hybrid imputation method: imputation of left-censored missing data (missing values ≥ 50% of number of replicas) was done using QRILC method, instead missed at random data (< 50% of replicas) were imputed using KNN method. Log2 transformed LFQ intensities were centred by Zscore normalisation method of Perseus (https://www.maxquant.org/perseus/) and then subjected to Student's Tests (S0 = 0.1, FDR<0.05) in order to discover Differentially Abundant Proteins (DAPs). Hierarchical clustering analysis was carried out using Perseus software and default parameters. DAPs categorisation was achieved using TAIR GO annotation tool (https://www.arabidopsis.org/tools/bulk/go/index.jsp). GO term enrichment analysis was performed using the PANTHER classification system (http://geneontology.org) [62].

## Results

### Depletion of *PRPS1* gene leads to embryo lethality

Previous studies reported on the Arabidopsis *prps1-1* knock-down mutant phenotype, characterised by pale-green leaves, reduced growth rate and photosynthetic performance, as result of hampered plastid protein synthesis [27,31]. However, the consequences of the complete inhibition of PRPS1 protein accumulation in Arabidopsis plastids has not yet been investigated. To fill this gap, the nuclear *PRPS1* gene was edited by targeting the first exon of *PRPS1* coding sequence using the CRISPR/Cas9 technology. Plant lines devoid of the Cas9 gene were obtained in T2 generation and the DNA region complementary to the designed guide RNA was sequenced (primers and guide RNA sequences are listed in S1 Table). Only *PRPS1/prps1* heterozygous plants, with wild-type-like plant size, leaf pigmentation and photosynthetic performance, could be isolated and the resulting *prps1-2* allele showed the deletion of a Cytosine in the first exon (+80 bp from the transcription starting site), right downstream the ATG translation start codon (+ 4 bp from ATG, Fig 1A and 1B). This event disrupts the *PRPS1* reading frame and introduces a premature Umber STOP codon in place of Leu-11 (+108 bp from the transcription starting site) (Fig 1A). Furthermore, only *PRPS1/prps1-2* heterozygous and *PRPS1/PRPS1* homozygous plants could be identified within the progeny of the self-fertilised *PRPS1/prps1-2* heterozygous line, showing the 2-to-1 mendelian segregation ratio typical of mutations causing embryo lethality, as in the case of other plastid ribosomal protein knock-out mutants [27,35,63]. Accordingly, the observation of *PRPS1/prps1-2* developing siliques at 10 Days After Fertilisation (DAF) revealed one-quarter of the seeds to be albino (Fig 1C), indicating that *PRPS1* is essential during early stages of embryogenesis and seed development. In particular, optical section of cleared, whole-mount seeds from *PRPS1/prps1-2* siliques at 3 DAF showed that around 25% of the embryos were arrested at the globular development stage, displaying a disorganised cell division pattern similar to the ones previously described for other knock-out mutants in essential plastid ribosomal proteins (Fig 1D) [27]. Furthermore, the defect in embryo development was fully rescued in *prps1-2 pPRPS1::PRPS1* plants, obtained by introducing the *PRPS1* genomic DNA into the *PRPS1/prps1-2* genetic background and isolating wild-type homozygous *prps1-2* plants carrying the *pPRPS1::PRPS1* construct (Fig 1C and 1D).

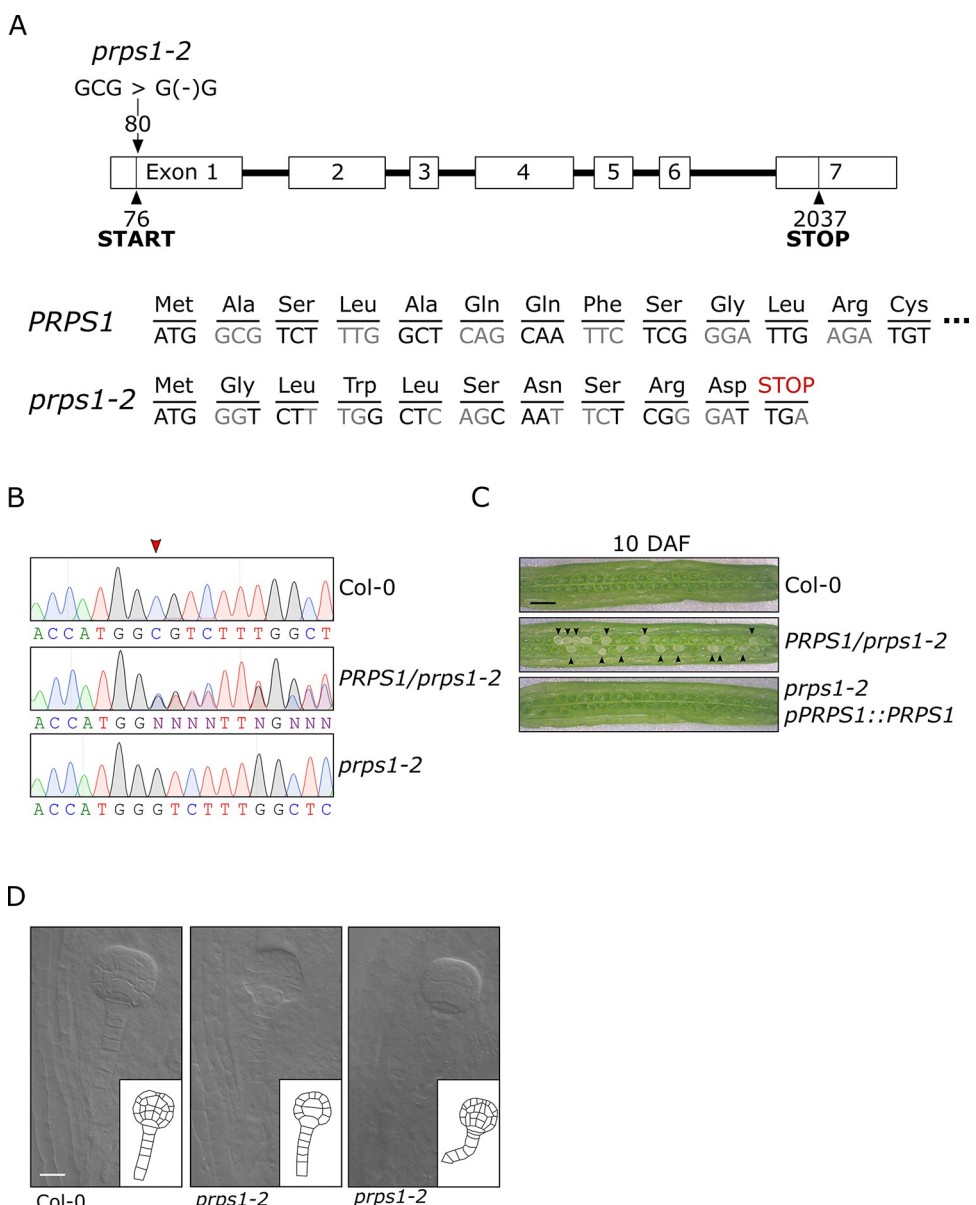

**Fig 1. Molecular description of *prps1-2* allele and the corresponding embryo-lethal phenotype.** A) Schematic representation of *PRPS1* gene. Exons are indicated as numbered white boxes, and introns as black lines. Positions of start and stop codons with respect to the transcription initiation site, as well as the CRISPR/Cas9-induced deletion (*prps1-2*), are indicated. The *PRPS1* reading frame and corresponding amino acids in wild-type and *prps1-2* mutant are also reported. B) Sequencing electropherogram showing the single nucleotide deletion (coding strand) in *prps1-2* hetero- and homozygous mutants with respect to Col-0. The red arrowhead indicates the position of the CRISPR/Cas9-induced deletion. "N" readings in *PRPS1/prps1-2* lines result from double peaks, generated by Cas9-induced deletion in one of the two homologous chromosomes. C) Morphological characteristics of developing seeds in siliques at 10 Days After Fertilization (DAF) of Col-0, heterozygous *PRPS1/prps1-2* and *prps1-2* complemented with *pPRPS1::PRPS1* genomic locus. Around one quarter of white seeds are clearly distinguishable among the green seeds of *PRPS1/prps1-2* siliques. Bars = 2 mm. D) Cleared whole mount of Col-0, heterozygous *PRPS1/prps1-2* and *prps1-2 pPRPS1::PRPS1* complemented seeds containing embryos at the globular stage (3 DAF). Bars = 20 μm.

## *PRPS1* over-expression impairs chloroplast activity and biogenesis

Besides the embryo lethal phenotype caused by the *prps1-2* allele and the slightly pale cotyledons and true leaves, together with hampered photosynthetic efficiency ($F_V/F_M$), typical of plants carrying the knock-down *prps1-1* allele (Fig 2A; see also [27,31]), the *CaMV35S*-mediated over-expression of *PRPS1* gene in the Arabidopsis Col-0 background (*oePRPS1*), also resulted in a visible phenotype, characterised by virescent young leaves with a marked drop in $F_V/F_M$ values (Fig 2A). Interestingly, *oePRPS1* seedlings were incapable of over-accumulating the PRPS1 protein (Fig 2B), showing an accumulation level lower than the one observed in *prps1-1* leaves, despite the *PRPS1* transcript level having been about two-fold the Col-0 control leaves (Fig 2C, see also [28]). On the contrary, Col-0 plants carrying the *CaMV35S::PRPL4* construct (*oePRPL4*), here used as the control, were able to accumulate up to 25–30 fold more *PRPL4* transcripts and almost double the amount of PRPL4 protein without affecting chloroplast biogenesis and activity, as shown by *oePRPL4* lines indistinguishable from Col-0 (Fig 2A–2C).

In order to investigate this aspect further, *PRPS1* and *PRPL4* coding sequences were cloned into the *pOp/LhG4* vector, which allows the inducible over-expression of the two genes once the glucocorticoid analogue dexamethasone (DEX) is provided. The two constructs were introduced into Arabidopsis Col-0 genetic background, via Agrobacterium-mediated transformation, resulting in the dexamethasone-inducible lines *indPRPS1* and *indPRPL4*. In the absence of DEX, the *indPRPS1* line was virtually indistinguishable from Col-0 when grown on MS medium under sterile conditions for 16 days (Fig 2A). Conversely, when the growth medium was supplemented with 2μM DEX, *indPRPS1* seedlings showed a leaf virescent phenotype and a drop in photosynthetic performance, together with a reduced accumulation of PRPS1 protein resembling the phenotype of *prps1-1* and *oePRPS1* lines (Fig 2A and 2B). This was despite *PRPS1* transcripts accumulated to levels higher than Col-0 control leaves (Fig 2C). On the other hand, the inducible overexpression of *PRPL4* resulted in an increased accumulation of *PRPL4* transcripts and protein, similar to *oePRPL4* seedlings (Fig 2B and 2C), without any impact on chloroplast activity and leaf greening (Fig 2A). Additional independent over-expressor and inducible lines for both *PRPS1* and *PRPL4* genes displayed comparable visible and molecular phenotypes, as shown in S1A–S1C Fig.

To understand how chloroplast ultrastructure organization is affected by the increased expression of *PRPS1* gene, thin sections of emerging young leaves from 16 DAS seedlings were observed under Transmission Electron Microscopy (TEM; Fig 3). As expected, Col-0, *prps1-1*, *indPRPS1* − DEX, *indPRPL4* ± DEX, and *oePRPL4* mesophyll cells displayed properly developed chloroplasts with the typical organization in grana stacks and stroma lamellae (Fig 3). However, both the induction (*indPRPS1* + DEX) and the constitutive increased expression of *PRPS1* caused the formation of miss-shaped and swollen chloroplasts containing enlarged plastoglobuli in the stroma (Fig 3). Furthermore, large budding vesicles with electron dense material were detectable, suggesting ongoing chloroplast degradation, resembling the fission-type ATG-independent micro-autophagy [34,51,52,64]. In some cases, entire round-shaped chloroplasts, detached from the plasma membranes and with still recognizable grana stacks, were observed inside the vacuole, compatible with the ATG-dependent micro-autophagy process (Fig 3J) [34,65]. To further prove that chloroplasts are indeed undergoing vacuole-mediated degradation, we investigated the relative expression of genes associated with either the chloroplast ATG-dependent or ATG-independent chloroplast quality-check and degradation pathways (S2 Fig). Strikingly, both *ATI1* [66] and *ATG8f* [67] transcripts were highly enriched in plantlets with increased *PRPS1* transcript accumulation, driven by either *CaMV35S*- or DEX-induced promoters (S2A Fig), whereas *NPC1* and *VPS15* [68] were the only genes of the

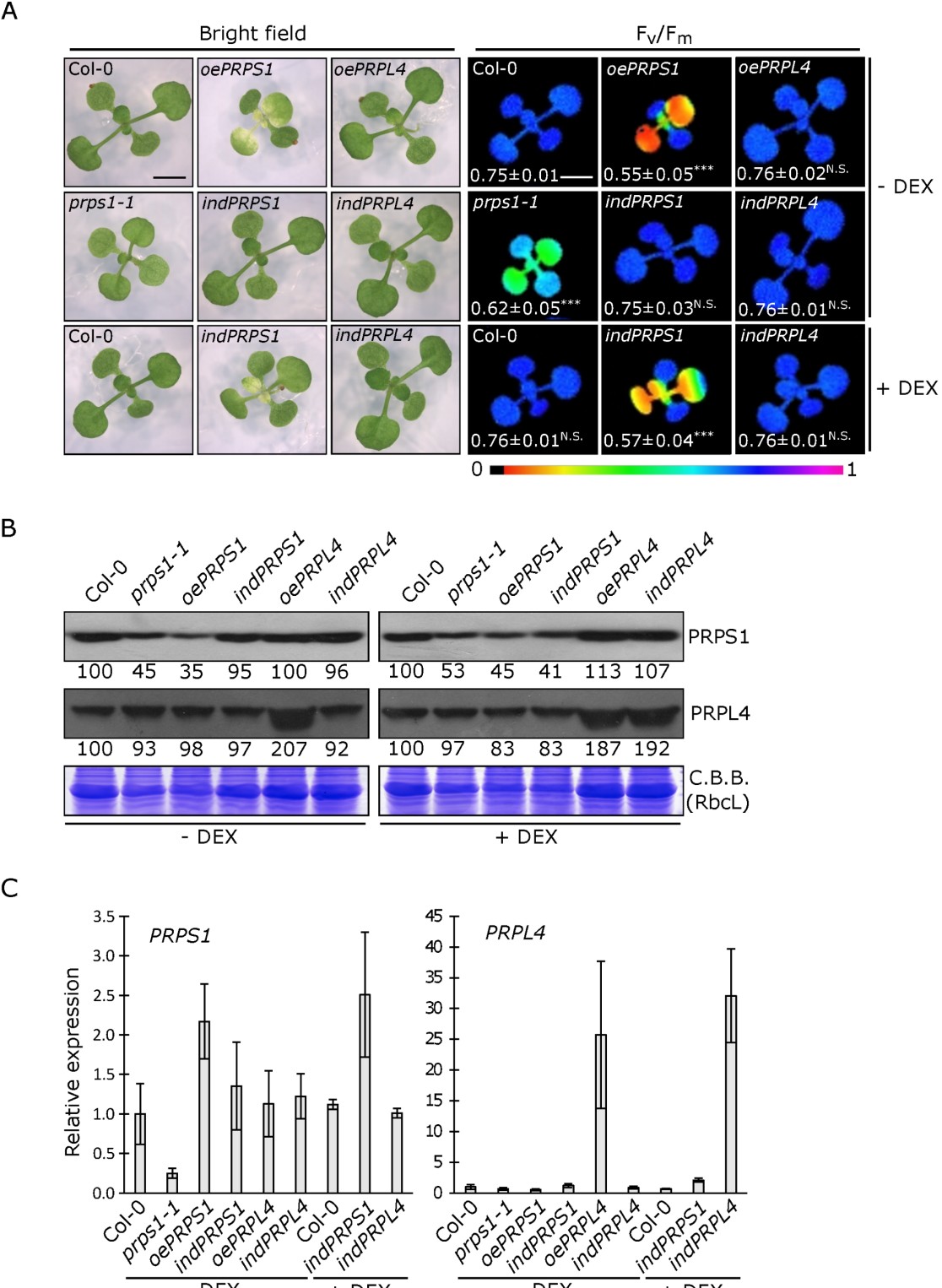

**Fig 2. Visible and molecular phenotypes of plantlets carrying altered amount of PRPS1 and PRPL4 proteins.** A) Visible phenotypes and photosynthetic efficiency ($F_V/F_M$) of 16 Days After Sowing (DAS) plantlets grown on MS medium devoid (- DEX) or supplemented (+ DEX) with 2 μM dexamethasone. On the left (bright field), the visible phenotypes shown by Col-0, the knockdown *prps1-1* mutant, *oePRPS1* and *oePRPL4* constitutive over-expressing lines, *indPRPL4* and *indPRPS1* inducible lines. On the right panel ($F_V/F_M$), the photosynthetic efficiency of each genotype is shown both as false colour imaging and average

values ± standard deviations. Statistical significance calculated via Student's t-test (*** indicates $P < 0.001$; N.S. not significant). Scale bar = 1 cm. B) Immunoblots of total protein extracts from Col-0, *prps1-1*, *oePRSP1*, *indPRPS1*, *oePRPL4* and *indPRPL4* 16 DAS plants grown on MS medium devoid (- DEX) or supplemented (+ DEX) with 2 μM dexamethasone. PRPS1 and PRPL4 specific antibodies were used for immuno-decoration. Coomassie Brilliant Blue (C.B.B.) stained gels are shown as loading control. Numbers show the relative protein abundance with respect to Col-0 (indicated as 100). Standard deviation was below ± 15%. One filter out of three biological replicates is shown. C) Relative expression values of *PRPS1* and *PRPL4* genes determined by qRT-PCR analyses of total RNA extracted from Col-0, *prps1-1*, *oePRSP1*, *indPRPS1*, *oePRPL4* and *indPRPL4* 16 DAS plants grown on MS medium devoid (- DEX) or supplemented (+ DEX) with 2 μM dexamethasone. Results of one out of three biological replicates are shown. Error bars indicate standard deviations of three technical replicates. Phenotypic and molecular data of other three independent transgenic lines for each genotype shown here are reported in S1 Fig.

ATG-independent pathway significantly up-regulated in *oePRPS1* plantlets (S2B Fig). On the contrary, transcripts of *SAG12* gene involved in senescence-associated vacuoles (SAV) pathway [69] could not be detected, in agreement with data from BAR Toronto ePlant [70], most probably as consequence of the fact that the SAV pathway is mainly activated during senescence rather than upon stress conditions.

## *PRPS1* short-term increased expression inhibits plastid protein translation

To gain a dynamic view on PRPS1 function, the kinetics of *PRPS1* transcript and protein accumulation was monitored in leaf discs (6 mm in diameter) from young leaves of *indPRPS1* plantlets infiltrated with 2 μM DEX. Leaf discs were sampled at 0, 3, 6 and 24 hours after infiltration (HAI) and transcript and protein accumulation were monitored by RT-qPCR (Fig 4A) and immunoblotting (Fig 4B), respectively. As control, the same experimental set-up was used to monitor the induction of *PRPL4* expression (Fig 4A and 4C). The accumulation of *PRPL4* and *PRPS1* mRNAs indicated that both inducible lines were able to specifically express the related genes with comparable kinetics and transcript amounts (Fig 4A). Both lines reacted to the presence of DEX showing a high expression level of the related transcripts at 3 HAI, reaching the peak at 6 HAI and a significant decrease at 24 HAI. Nevertheless, the induction of *PRPS1* expression failed to yield the over-accumulation of PRPS1 protein. Indeed, PRPS1 protein level remained stable until 3 HAI, while diminished to almost undetectable levels from 6 to 24 HAI (Fig 4B). Interestingly, other plastid ribosomal proteins, such as PRPL4 and PRPS5, showed a marked decreased over time, indicating a general alteration of plastid ribosome accumulation, and possibly of plastid translation. On the contrary, the 24-hour induction of *PRPL4* resulted in more than two-fold accumulation of PRPL4 protein with respect to time 0 (Fig 4C). In particular, the increase in PRPL4 protein accumulation was observed over time, starting from 3 HAI and reaching the largest amount at 6 HAI, while the accumulation of PRPS1 and PRPS5 plastid ribosomal proteins remained unaltered.

To investigate the possible negative effect of *PRPS1* inducible expression on plastid protein translation, *indPRPS1* leaf discs were incubated in MS medium (± DEX) for 6 hours and then infiltrated with [35]S-Methionine and cycloheximide, allowing for the detection of *de novo* synthesized plastid-encoded proteins, while blocking the cytosolic translation. As shown by the pulse-labelling experiment, the synthesis rate of RbcL and D1/D2 proteins were markedly reduced in *indPRPS1* + DEX, over 15 and 30 minutes, when compared to *indPRPS1* leaf discs in the absence of dexamethasone (- DEX; Fig 4D), proving that *PRPS1* inducible over-expression leads to plastid translation inhibition. This aspect was investigated further by isolating the PRPS1-containing complexes and monitoring the PRPS1-to-ribosome stoichiometry and the accumulation of "free" PRPS1 fraction, as previously reported in *E. coli* [16]. To do so, *indPRPS1* leaf material (6 hours ± DEX) was subjected to sucrose gradient fractionation and probed with the PRPS1 antibody (Fig 4E). As observed in Chlamydomonas [50], PRPS1 protein was found in two distinct populations, as "Low Molecular Weight (LMW, free PRPS1

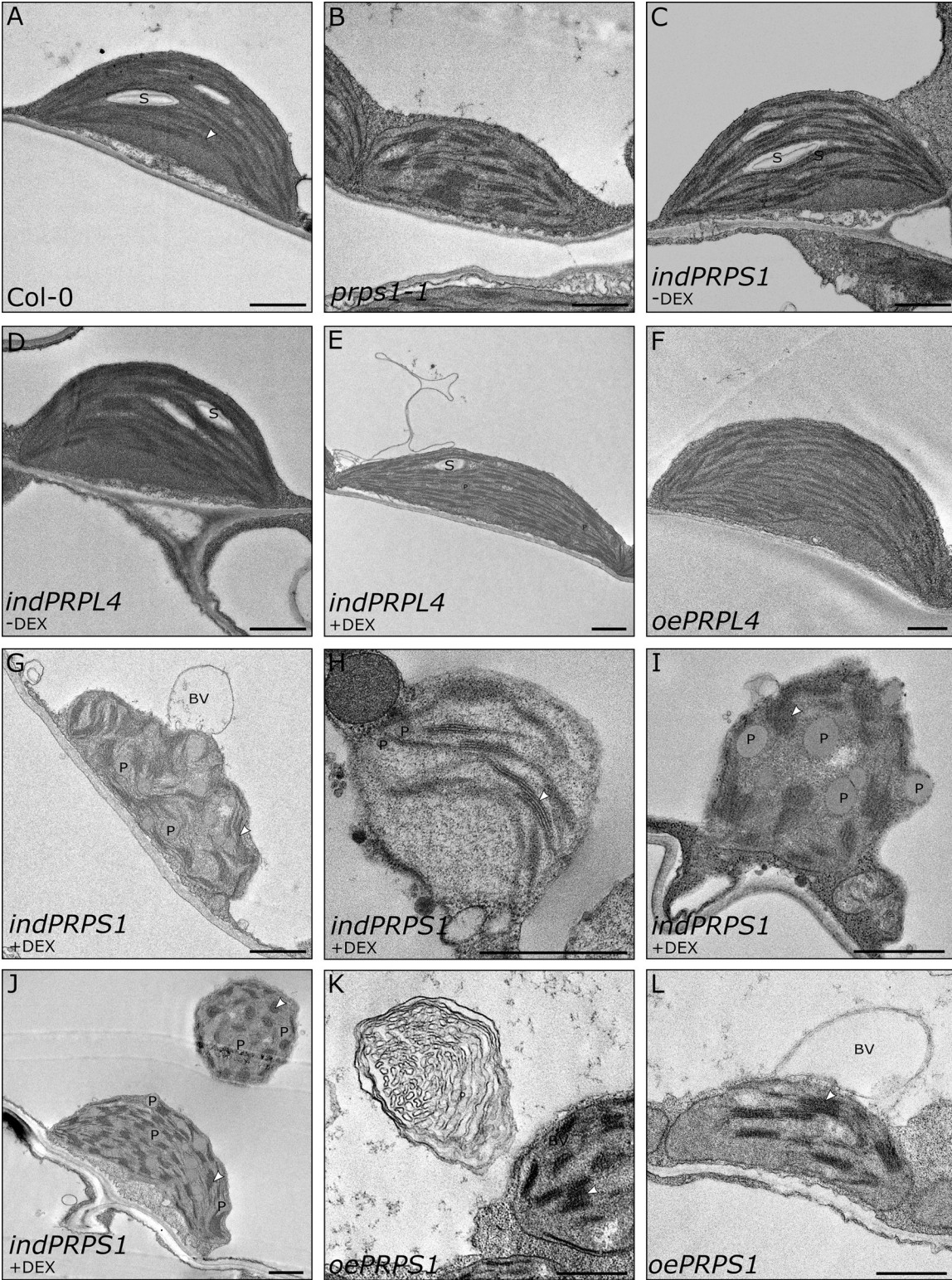

**Fig 3.** TEM micrographs of chloroplasts in mesophyll cells of Col-0 (A) control, *prps1-1* (B), *indPRPS1* (C, G, H, I, J), *indPRPL4* (D, E), *oePRPL4* (F) and *oePRPS1* (K, L) leaves obtained from 16 DAS plantlets grown on MS medium supplemented, or not, with DEX. The young portion of leaves, proximal to the petiole, showing the yellow to pale-green phenotype in *oePRPS1* and *indPRPS1* lines, was used for the analyses. S: starch granules; White arrowhead: thylakoid membranes; BV budding vesicles; P: Plastoglobules.

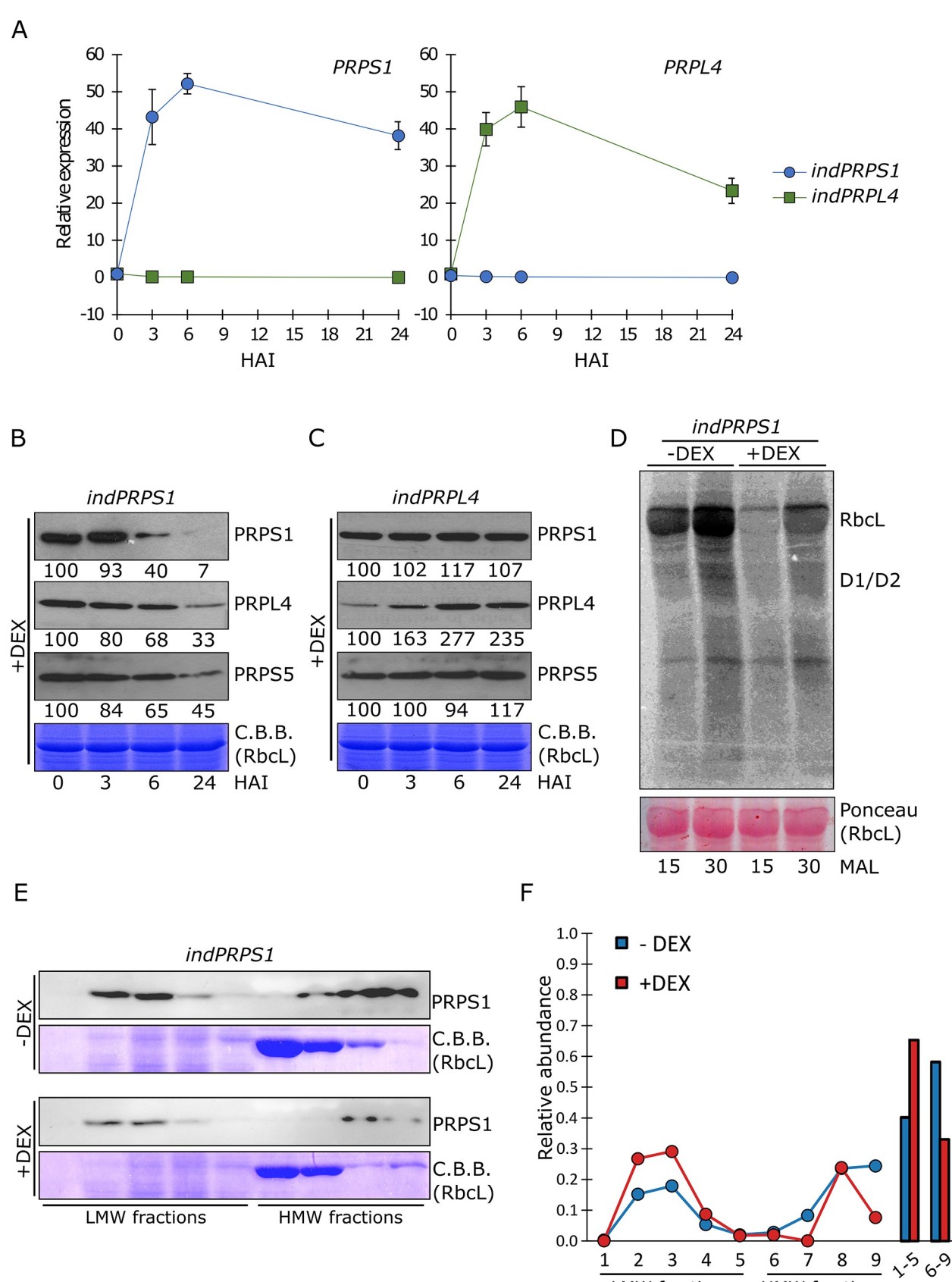

**Fig 4. Molecular features of *indPRPS1* and *indPRPL4* lines upon DEX-induced gene expression.** A) Relative expression values of *PRPS1* and *PRPL4* genes determined by qRT-PCR analyses of total RNA extracted from *indPRPS1* (circles, blue) and *indPRPL4* (squares, green) leaf discs sampled at 0, 3, 6 and 24 Hours After Infiltration (HAI) with 2 μM dexamethasone (DEX). Data from one out of three biological replicates are shown. Error bars indicate standard deviations of three technical replicates. B) Immunoblots of total protein extracts from *indPRPS1* leaf discs sampled at 0, 3, 6 and 24 HAI with 2 μM DEX. Filters were incubated with antibodies raised against PRPS1, PRPL4 and PRPS5 plastid ribosomal proteins. C.B.B.-stained gels are shown as loading control. Numbers show the relative protein abundance with respect to 0 HAI (indicated as 100). Data from one out of three biological replicates are shown. Standard deviation was below 15%. C) Immunoblots of total protein extracts from *indPRPS4* leaf discs, sampled and analysed as in B, used as control. D) Total protein extracts from Col-0 and *indPRPS1* leaf discs previously incubated in DEX-containing MS medium for 6 hours and then infiltrated with $^{35}$S-Methionine in the presence of cycloheximide, an inhibitor of cytosolic protein synthesis. The leaf proteins were sampled 15 and 30 minutes after labelling (MAL). RbcL and D1/D2 proteins signals are indicated. The ponceau-stained filter is shown as loading control. E) *indPRPS1* ± DEX (6 HAI) leaf material was subjected to sucrose gradient fractionation aimed to isolate PRPS1-containing complexes. Filters were immuno-decorated with PRPS1 antibody to show the accumulation of PRPS1 in Low Molecular Weight (LMW) and High Molecular Weight (HMW) fractions. C.B.B.-stained gels are shown as loading control. F) Quantification of PRPS1 accumulation in LMW (1–5) and HMW (6–9) fractions as evaluated by Image Lab software on representative blots. Each lane has been quantified as absolute value (see also Materials and Methods). The sum of all signals has been set as 1. The values reported in the graph are relative to the total sum for each condition (DEX -/+).

fraction)", bound solely to the mRNA, and as "High Molecular Weight (HMW)", corresponding to the S1 fraction bound to the ribosome core. The increase of PRPS1 presence in the LMW fractions of *indPRPS1* + DEX line, compared to the − DEX counterpart (Fig 4E and 4F), supports further the inhibition of plastid translation observed in Fig 4D. Consistent with these data, we observed that PRSP1 was almost absent in HMW fraction in *oePRPS1* samples (S3 Fig).

## The over-accumulation of Arabidopsis PRPS1 protein inhibits *Escherichia coli* cell growth

Similarly to Arabidopsis, the depletion of the ribosomal protein S1 (Rps1) in *E. coli* cells leads to lethality [11]. Moreover, the down-regulation as well as the over-accumulation of Rps1 impair protein translation by altering the stoichiometry between Rps1 and the ribosome core, leading to bacteriostatic effects ([16,33]; see also Fig 5). The *E. coli* S1 protein is 557 aa long with a molecular weight of 61.2 kDa and possesses six S1 domains ([18]; Fig 5A) that are typical of several RNA binding proteins [71–73]. In *A. thaliana*, the S1 homologous PRPS1 is markedly smaller, with 373 aa and a molecular weight of 40.5 kDa, as mature form. The *in silico* analysis of PRPS1 amino acid sequence identified three S1 domains (Fig 5A), in agreement with early analyses of spinach S1 protein [74]. In addition, PRPS1 protein shows a high degree of identity (*i.e.* about 50%) with the S1 ribosomal proteins from cyanobacteria, which possess three S1 domains as well [19,75]. In order to investigate the possible relationships between the three S1 domains identified in PRPS1 and the six S1 domains in *E. coli* S1, the amino acidic sequences of each S1 domain were aligned and clustered in a phylogenetic tree (S4 Fig). The resulting tree showed that domains 1 and 2 of PRPS1 are more similar to the corresponding domains of S1, while the PRPS1 domain 3 clusters together with the domains 3, 4 and 5 of S1.

The possible functional homology between the two proteins was then investigated by introducing the *PRPS1* gene in *E. coli* cells, for over-expression and complementation assays. To experimentally test the ability of PRPS1 to complement S1 functions and to repress cell growth when over-accumulated in bacterial cells, both *rpsA* (encoding the S1 protein) and *PRPS1* coding sequences were cloned into pQE31-pREP4 plasmid system under the control of *pT5-lacO* promoter in pQE31. The plasmids were then introduced into the arabinose-dependent strain C-5699 [33], in which the chromosomal *rpsA* gene is transcribed from the *araBp* promoter, generating *indrpsA* and *indPRPS1* strains. Such systems provide a mechanism to deplete cells of the endogenous S1 protein in absence of arabinose and presence of glucose [16,33] and to modulate the expression of either *rpsA* or *PRPS1* genes cloned in pQE31 depending on IPTG concentration.

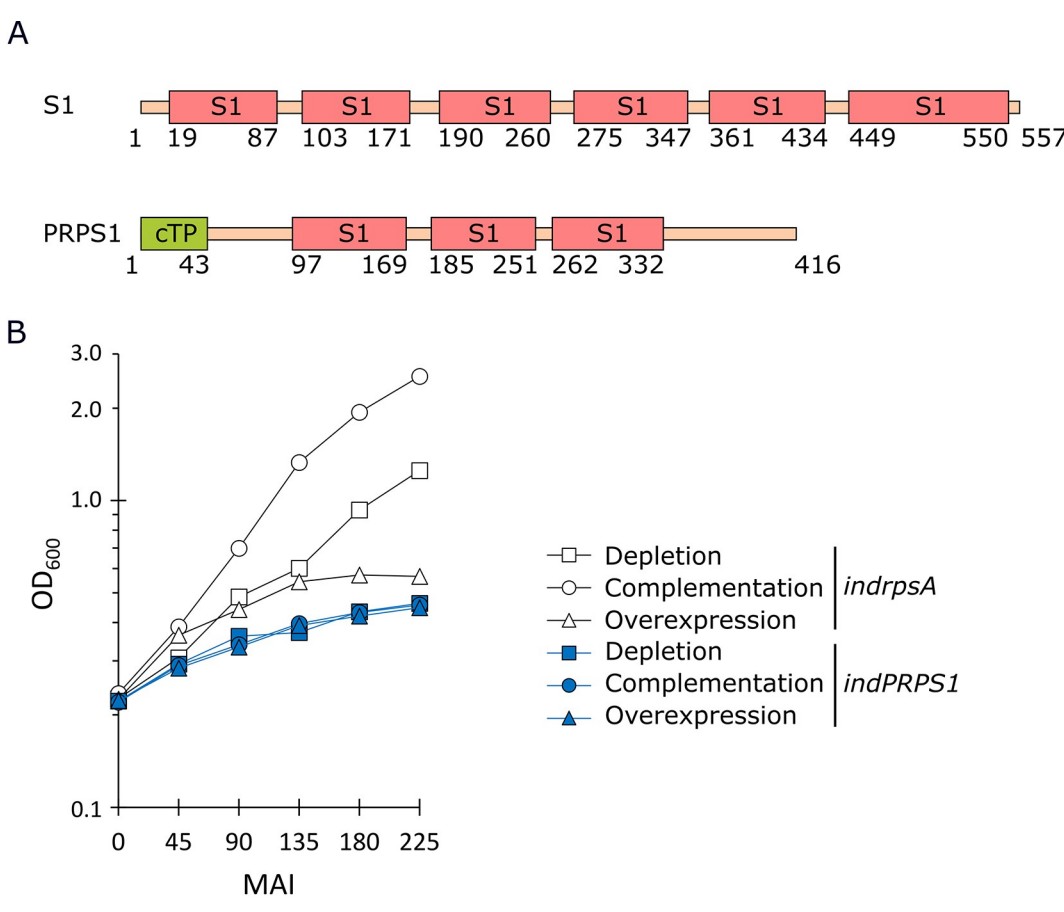

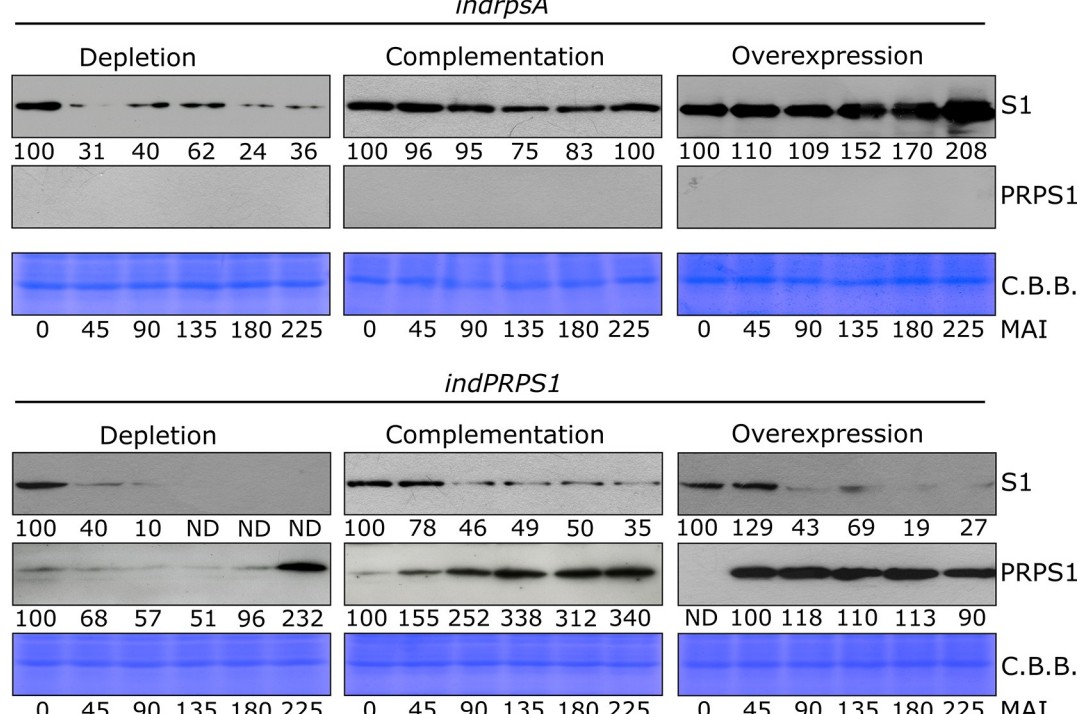

**Fig 5. Comparison of PRPS1 and S1 protein activities in *E. coli* cells.** A) Schematic representation of S1 and PRPS1 proteins. S1 domains are depicted as pink boxes, while the chloroplast transit peptide (cTP) of PRPS1 protein as green box. Numbers indicate the position of amino acid residues. Domain predictions are based on InterPro online tool (https://www.ebi.ac.uk/interpro/). B) OD$_{600}$ measurements of *E. coli indrpsA* (white symbols) and *indPRPS1* (blue symbols) strains grown under Depletion (0.4% glucose; squares), Complementation (0.4% glucose + 0.01 mM IPTG; triangles) or Overexpression (1% arabinose + 1 mM IPTG; circles) conditions. Strains were sampled at 0, 45, 90, 135, 180 and 225 Minutes After Induction (MAI). Data from one out of three biological replicates are reported. C) Immunoblots of total protein extracts from *indrpsA* and *indPRPS1 E. coli* strains, grown as described above, sampled at 0, 45, 90, 135, 180 and 225 MAI, using S1 and PRPS1 specific antibodies. C.B.B.-stained gels are added as loading controls. Numbers show the relative protein abundance with respect to 0 MAI. Data from one out of three biological replicates are shown. Standard deviation was below 15%.

Both *indrpsA* and *indPRPS1* strains were cultured in three different conditions: Depletion (0.4% glucose and no IPTG), Complementation (0.4% glucose and 0.01 mM IPTG) and Overexpression (1% arabinose and 1 mM IPTG). The growth conditions for the complementation assay were experimentally optimized based on the growth of *indrpsA E. coli* strain. Cell growth was measured every 45 minutes up to 225 minutes after the induction (MAI, Fig 5B). At each time point cells were sampled, normalized on the optical density (OD$_{600}$) value, and the total protein extract was used to detect the accumulation of S1 and PRPS1 proteins in both strains (Fig 5C). Under Depletion conditions, both strains showed almost completely impaired growth rate, due to the limited accumulation/absence of S1 protein. When grown in Complementation conditions, *indrpsA* cells were able to actively replicate and showed comparable amounts of S1 protein at each time point, indicating that the experimental conditions were properly set up to induce a wild-type-like S1 protein accumulation. On the other hand, *indPRPS1* cells failed to sustain growth in such conditions despite the gradual accumulation of PRPS1 protein, proving the inability of PRPS1 to complement the S1 function in *E. coli*. As expected, under Overexpression conditions, *indrpsA* strain growth was inhibited due to the excessive amount of S1 protein (Fig 5B and 5C). Strikingly, the over-accumulation of PRPS1 protein was effectively repressing the bacterial growth and led to S1 protein depletion, too (Fig 5B and 5C). It is worth noting, that the overaccumulation of PRPS1 Arabidopsis protein was able to repress the growth of *E. coli* cells, comparably to S1 overaccumulation, even when the overexpression of either *rpsA* or *PRPS1* genes was achieved in the C-1a *E. coli* strain, devoid of the conditional depletion system of the endogenous S1 protein (S5A Fig). To better understand whether PRPS1 protein is capable of interacting with *E. coli* ribosomes under overexpression conditions, we sampled both *indrpsA* and *indPRPS1* cells at 90 MAI. Cell lysates were then fractionated into ribosome-unbound (supernatant, SN) and ribosome-bound (pellet, P) fractions and analysed via immunoblotting to detect either S1 or PRPS1 protein localisation. Interestingly, PRPS1 was retrieved in both ribosome-bound and -unbound fractions, similarly to S1 from *E. coli*, suggesting that the Arabidopsis PRPS1 can compete for the ribosome core with *E. coli* S1 protein (S5B Fig). Taken together, these data indicate that PRPS1 protein is able to inhibit *E. coli* growth when over-expressed in addition to the endogenous S1, whereas it is unable to functionally replace the *E. coli* endogenous S1 protein.

## PRPS1 accumulation is negatively regulated by chloroplast CLP protease complex

The phenotypes observed in Arabidopsis plants upon *PRPS1* overexpression indicate that the abundance of PRPS1 protein must be kept under a strict post-transcriptional control to prevent inhibition of protein synthesis and chloroplast damage (see Figs 2–4). This notion is supported further by the inhibitory role of PRPS1 protein over-accumulation on *E. coli* cell growth (Figs 5 and S5). In order to investigate the molecular mechanism responsible for controlling PRPS1 protein abundance, leaf discs were infiltrated with DEX supplemented with

lincomycin (LIN), a specific inhibitor of plastid 70S ribosomes. Strikingly, a large accumulation of PRPS1 protein, most probably as result of PRPS1 degradation suppression, could be observed even after 24 hours from DEX infiltration, unlike the control sample (Fig 6A). Intriguingly, the most relevant chloroplast stromal protease is represented by the CLP complex, which is composed by several nuclear-encoded subunits and one plastid-encoded component, ClpP1, that is part of central proteolytic core [39,76]. To investigate whether the CLP complex could indeed be responsible for maintaining PRPS1 protein below levels that would otherwise cause damages to the chloroplast, we crossed *prps1-1* with mutants altered either in chloroplast protein translation, *prpl11-1*, or lacking two plastid chaperones required to feed CLP protease with protein substrates, *clpc1-1* and *clpd-1* ([44]; Fig 6B). *prps1-1 prpl11-1* double mutant showed reduced growth rate (S6 Fig) and a slight decrease in photosynthetic efficiency ($F_V/F_M$) with respect to *prpl11-1* parental line. Similar genetic interactions were observed in *prps1-1 clpc1-1* double mutant, which showed a severe reduction in growth rate with respect to *prps1-1* and *clpc1-1* single mutants. Interestingly, *prps1-1 clpd-1* double mutant showed partially restored growth rate, close to wild-type-like levels, and a slight recovery of $F_V/F_M$ parameter. Strikingly, the accumulation of PRPS1 protein increased about two-fold in all the double mutants tested (Fig 6C) with respect to *prps1-1*, further supporting the notion that plastid translation and the CLP protease complex play an important role in controlling PRPS1 abundance in the stroma of chloroplasts. To further verify the role of CLP protease complex in the regulation of PRPS1 protein abundance in plastids, the *clpc1-1* and *clpd-1* mutants were manually crossed with *indPRPS1* lines to yield the *indPRPS1 clpc1-1* and *indPRPS1 clpd-1* genotypes. Leaf discs harvested from these lines were then infiltrated with DEX and PRPS1 accumulation was monitored over 24 hours. Importantly, PRPS1 protein was able to accumulate in large amount upon DEX induction in both lines altered in CLP activity, unlike in *indPRPS1* leaf discs (Fig 6D). Similar results were observed in the crosses obtained with *indPRPS1 #2* line (S7A Fig). Additionally, *clpc1-1* and *clpd-1* mutants were also crossed with *oePRPS1* lines. Interestingly, the obtained *oePRPS1 clpc1-1* and *oePRPS1 clpd-1* lines sensibly recovered the photosynthetic phenotype of *oePRPS1* (S7B Fig), most probably as a consequence of the increased PRPS1 abundance (S7C Fig), further supporting the notion that the CLP protease complex plays an important role in controlling PRPS1 abundance in the stroma of chloroplasts. Finally, no accumulation of PRPS1 precursors were detected outside the chloroplasts under overexpression conditions by immunoblot analyses on plastid-enriched fraction and soluble extra-plastid fraction isolated from Col-0, *prps1-1* and *oePRPS1* 16 DAS plantlets grown on soil, indicating that the reduction of PRPS1 protein upon its overexpression is mainly due to the activity of CLP protease (S8 Fig).

## Short-term induction of *PRPS1* and *PRPL4* gene expression induces different nuclear gene expression responses

The opposite behaviour of *indPRPS1* and *indPRPL4* plants in terms of protein pattern accumulation upon DEX-mediated induction of gene expression make them the ideal genetic material to investigate the primary nuclear gene expression response to changes in plastid ribosomal protein content. To this aim, a transcriptome analysis was performed on leaf discs harvested from *indPRPS1* and *indPRPL4* and vacuum infiltrated in either the absence or presence of DEX. In particular, total leaf RNA was extracted after 6 hours from infiltration, i.e., at the stage of maximal accumulation of *PRPS1* and *PRPL4* transcripts (see Fig 4A) and at the beginning of evident changes in protein pattern accumulation (see Fig 4B and 4C), but without major visible and photosynthetic defects, reducing at minimum the consequences of pleiotropic effects (S9 Fig), and subjected to Illumina sequencing. Principal component analyses (PCA) of

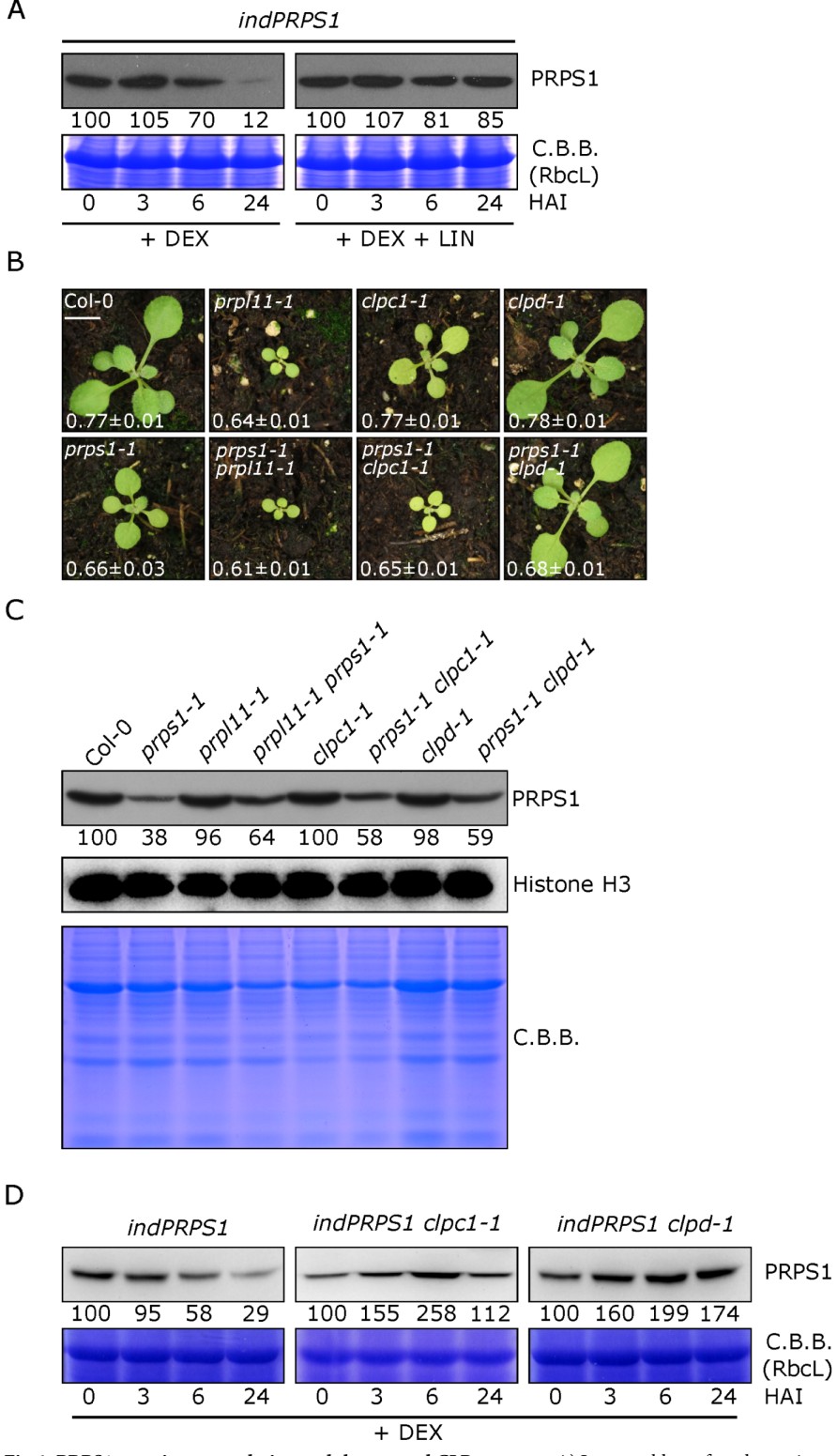

**Fig 6. PRPS1 protein accumulation and the stromal CLP protease.** A) Immunoblots of total protein extracts from *indPRPS1* leaf discs sampled at 0, 3, 6 and 24 HAI with 2 μM dexamethasone alone (+ DEX) or supplemented with 550 μM lincomycin (+ DEX + LIN). Filters were incubated with antibodies raised against PRPS1 protein. C.B.B.-stained gels are shown as loading control. Numbers below the immunoblots show the relative protein abundance with respect to 0 HAI (indicated as 100). Data are derived from one out of three biological replicates. Standard deviation

was below 15%. B) Visible phenotypes of Col-0 and *prps1-1*, *prpl11-1*, *clpc1-1*, *clpd-1*, *prps1-1 prpl11-1*, *prps1-1 clpc1-1* and *prps1-1 clpd-1* 16 DAS seedlings grown on soil and their photosynthetic efficiency (*Fv/Fm*). Average values ± standard deviations. Scale bar = 1 cm C) Immunoblot of total protein extracts from Col-0 and *prps1-1*, *prpl11-1*, *clpc1-1*, *clpd-1*, *prps1-1 prpl11-1*, *prps1-1 clpc1-1* and *prps1-1 clpd-1* mutants obtained by using a PRPS1 specific antibody. The immunoblot incubated with Histone H3 specific antibody and a C.B.B.-stained gel are shown as loading control. Numbers below the immunoblot show the relative protein abundance with respect to Col-0. Standard deviation was below 15%. D) Immunoblots of total protein extracts from *indPRPS1*, *indPRPS1 clpc1-1* and *indPRPS1 clpd-1* leaf discs sampled at 0, 3, 6 and 24 HAI with 2 µM DEX. Filters were incubated with antibodies raised against PRPS1 protein. C.B.B.-stained gels are shown as loading control. Numbers below the immunoblots show the relative protein abundance with respect to 0 HAI (indicated as 100). Data are derived from one out of three biological replicates. Standard deviation was below 15%.

*indPRPS1* and *indPRPL4* samples showed a clear separation between the untreated and treated samples (S10 Fig), despite the short induction time. The EdgeR package was used to identify differentially expressed genes (DEGs, listed in S2 Table), obtained by comparing data from *indPRPS1*-DEX with *indPRPS1*+DEX, and *indPRPL4*-DEX with *indPRPL4*+DEX (Fig 7A). Overall, 431 DEGs in *indPRPS1* upon DEX induction and 328 in *indPRPL4* were identified. Among them, 124 DEGs were in common between the two datasets. Next, the obtained DEGs were divided into up and down regulated genes both in *indPRPS1* and *indPRPL4* datasets, obtaining *indPRPS1*UP, *indPRPL4*UP, *indPRPS1*DOWN and *indPRPL4*DOWN lists, which were further compared to isolate unique DEGs in each group (Fig 7B, see also S2 Table). As a result, among the up-regulated DEGs, 161 were uniquely found in *indPRPS1* (Sheet A in S2 Table) and 159 DEGs in *indPRPL4* (Sheet B in S2 Table), while 107 DEGs were up-regulated in both lines (Sheet C in S2 Table). Among the down-regulated DEGs, 146 were unique for *indPRPS1* line (S2 D), 45 unique DEGs for *indPRPL4* (Sheet E in S2 Table) and only 14 were commonly down-regulated (Sheet F in S2 Table). Furthermore, among the 149 down-regulated DEGs in *indPRPS1*, 3 were upregulated in *indPRPL4* (Sheet G in S2 Table).

To investigate the biological functions activated or repressed by the overexpression of either *PRPS1* or *PRPL4* genes, the unique DEGs found up- or down-regulated in *indPRPS1* or *indPRPL4* were analysed with agriGO v2.0 online tool [56]. Next, further analysis by REVIGO [57] was preformed to retrieve Biological Process Gene Ontology terms and group them into functional categories (Fig 8 and S3–S5 Tables). The 161 DEGs up-regulated in *indPRPS1* produced strongly enriched GO terms associated with responses to endogenous factors or involvement in flowering and seeds production, such as "photoperiodism, flowering" (GO:0048573), "response to karrikin" (GO:0080167), "vegetative to reproductive phase transition of meristem" (GO:0010228) or "cellular response to hormone stimulus" (GO:0032870) (Fig 8A; see also S3 Table). On the other hand, the repressed biological functions found in *indPRPS1* were related to the production of secondary metabolites, defence against herbivores and oxidative stress such as "glucosinolate biosynthetic process" (GO:0019761), "sulfur compound biosynthetic process" (GO:0044272), "cell redox homeostasis" (GO:0045454) and "response to wounding" (GO:0009611), to cite a few of them (Fig 8B; see also S3 Table). Interestingly, the high enrichment in GO terms associated with flowering and reproductive phase transition is in good agreement with the anticipated flowering phenotype, calculated as number of rosette leaves at bolting, observed when *PRPS1* transcripts are overexpressed (Fig 9). In particular, in the case of plants cultivated on soil under growth chamber conditions, Arabidopsis Col-0 bolted with an average of about 10–11 rosette leaves while the *oePRPS1* line showed an anticipated flowering time, bolting at the stage of 8 rosette leaves, comparable to *prps1-1* behaviour (Fig 9A and 9B). The same anticipated flowering time occurred under short day growth conditions, with the wild-type bolting at 54 leaves while the overexpressing line flowered at 37 rosette leaves on average (Fig 9C). A similar observation was made when *indPRPS1* line was

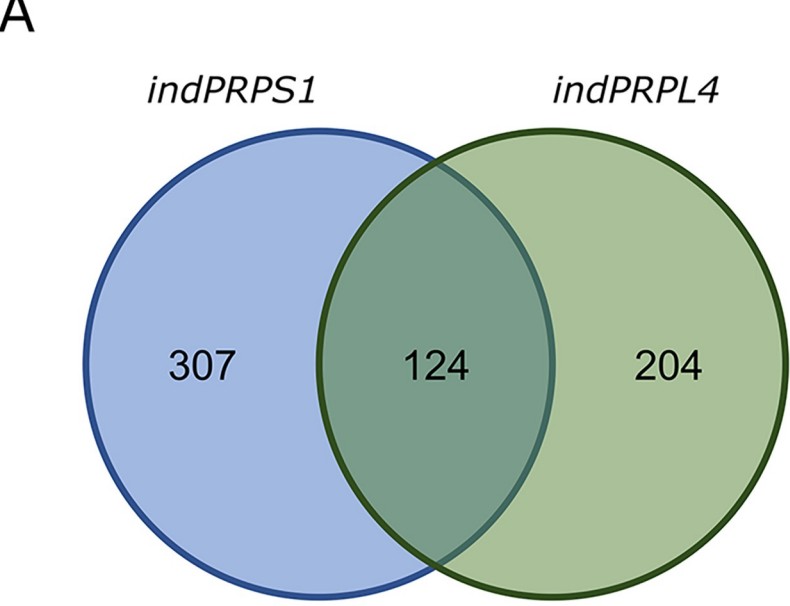

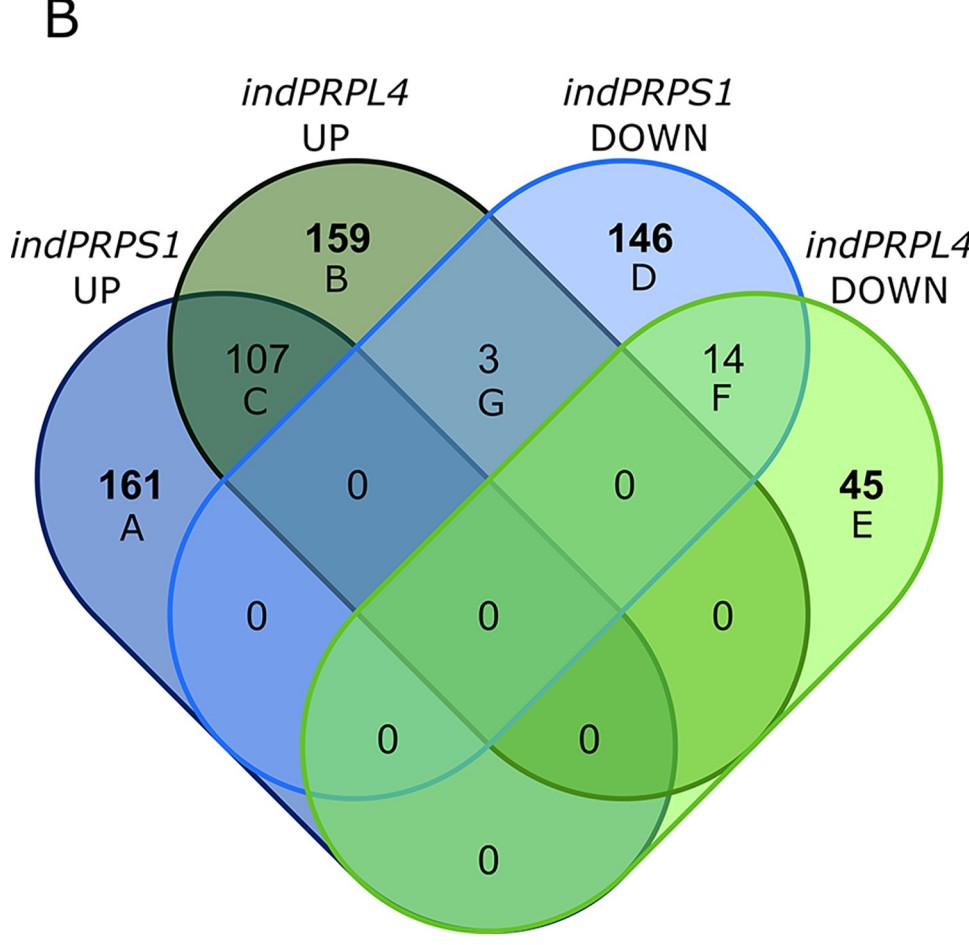

**Fig 7. Comparison of RNAseq transcriptomes from *indPRPS1* and *indPRPL4* lines infiltrated or not with DEX.**
A) Venn diagram showing the number of DEGs found in *indPRPS1* and *indPRPL4* upon induction. B) Venn diagram showing the distribution of DEGs resulted from the comparison of *indPRPS1*UP, *indPRPL4*UP, *indPRPS1*DOWN and *indPRPL4*DOWN lists. Unique DEGs in each group are in bold. Group A: Unique up-regulated DEGs found in *indPRPS1*; Group B: Unique up-regulated DEGs found in *indPRPL4*; Group C: Common up-regulated DEGs found in *indPRPS1* and *indPRPL4*; Group D: Unique down-regulated DEGs found in *indPRPS1*; Group E: Unique down-regulated DEGs found in *indPRPL4*; Group F: Common down-regulated DEGs found in *indPRPS1* and *indPRPL4*; Group G: up-regulated DEGs found in *indPRPL4* but down-regulated in *indPRPS1*.

grown on MS medium supplemented with 1% (w/v) sucrose and DEX, in which its number of leaves at bolting was 5, similarly to *oePRPS1* (Fig 9D). Instead, when *indPRPS1* was grown on MS medium without DEX, the number of leaves at bolting was about 6–7, comparable to what observed in *indPRPL4* and Col-0 plants.

Conversely, the analysis of the activated biological processes in *indPRPL4* showed a strong enrichment in GO terms associated with abiotic stress responses and protein homeostasis such as "response to hydrogen peroxide" (GO:0042542), "response to high light intensity" (GO:0009644), "response to heat" (GO:0009408), "response to oxidative stress" (GO:0006979) and "protein folding" (GO:0006457) (Fig 8C; see also S5 Table). These activated cellular pathways are in agreement with the observed increased levels of CLPB3, HSP90-1 and HSC70-4 proteins in *indPRPL4* line upon induction with DEX (see S11 Fig). In particular, CLPB3 protein abundance and even more the abundance of HSP90-1 and HSC70-4 proteins largely increased already at 3 HAI in *indPRPL4* leaf disks, unlike in *indPRPS1* samples, possibly generated by the overload of the plastid folding machinery caused by PRPL4 abundance and the consequent activation of chloroplast UPR. Interestingly, the unique down-regulated DEGs found in *indPRPL4* leaf disks resulted in no significantly enriched GO terms.

The RNA-seq library was prepared to enrich the PolyA-containing transcripts only, therefore, no plastid-encoded genes are represented. Therefore, to evaluate if the plastid transcription was affected by the DEX-mediated induction of either *PRPL4* or *PRPS1* genes, the very same samples used for transcriptome analyses were analysed via RT-qPCR to quantify the expression of a subset of PEP- and NEP-dependent plastid-encoded genes (S12 Fig). Interestingly, no significant differences in the expression of plastome genes in both lines after DEX treatment could be detected.

## Short-term induction of *PRPS1* and *PRPL4* gene expression induces different changes in proteomic profiles

To further investigate the molecular responses following the DEX-mediated induction of *PRPS1* and *PRPL4* genes, a proteomic analysis was performed on leaf tissue harvested from *indPRPS1* and *indPRPL4* plants and vacuum infiltrated either in the absence or presence of DEX and sampled, as in the case of transcriptome analysis, after 6 hours of induction. PCA analysis revealed that dexamethasone-mediated induction led to significant changes also at proteomic level (S13 Fig). Comparing induced (+DEX) and control groups (-DEX), the abundances of 58 and 77 proteins were altered in *indPRPS1* and *indPRPL4* samples, respectively (T Student's test, FDR < 0.05; listed in S6 Table). The minimal overlapping between the two datasets (only two common proteins) confirms the different cellular responses following the induction of *PRPS1* and *PRPL4* genes (Fig 10A). In particular, 27 and 31 proteins were up- and down-accumulated in *indPRPS1* leaf discs, whereas 45 and 32 proteins were up- and down-accumulated in *indPRPL4* samples, respectively (Fig 10B).

GO enrichment analysis of differentially abundant proteins (DAPs) found in *indPRPS1* samples revealed over-represented GO terms only among the down-accumulated proteins,

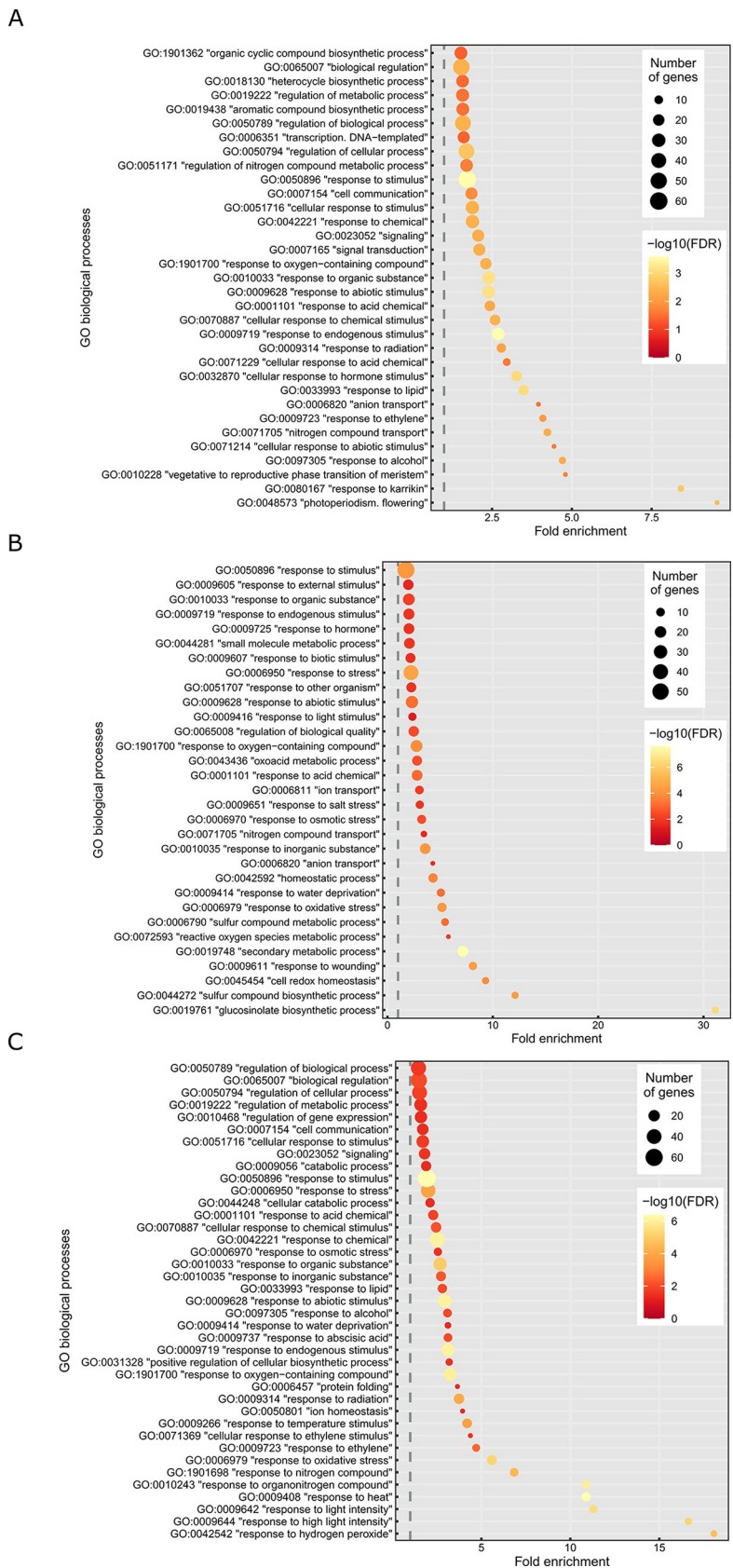

**Fig 8. Analysis of the biological functions activated or repressed by the short-term overexpression of *PRPS1* and *PRPL4* genes.** Significantly enriched GO terms identified through agriGO v2.0 and REVIGO online tools retrieved from A) unique up-regulated DEGs found in *indPRPS1*; B) unique down-regulated DEGs found in *indPRPS1;* C) unique up-regulated DEGs found in *indPRPL4*.

probably due to the small size of the data set. The enriched Biological Process terms were "translational elongation" (GO:0009658) and "chloroplast organization" (GO:0006414) (S7 Table). Among the enriched Cellular Component terms, "ribosome-associated quality control (RQC) complex" (GO:1990112), "transcriptionally active chromatin" (GO:0035327) and "chloroplast stroma" (GO:0009570) were found (S8 Table). All DAPs were grouped based on their GO annotations (Fig 11). Proteins up-regulated in response to *PRPS1* induction were mainly involved in "anatomical structure development" (GO:0048856), "catabolic process" (GO:0009056) and "response to light stimulus" (GO:0009416) (Fig 11A). In addition, the up-accumulated proteins in *indPRPS1* plantlets were mainly located in "nucleus" (GO:0005634) and "cytoplasm" (GO:0005737) (Fig 11B). Down-regulated DAPs produced GO annotations such as "biosynthetic processes" (GO:0009058), "RNA binding" (GO:0003729), "translation" (GO:0006412) and "post-embryonic development" (GO:0009791) (Fig 11A). Among down-regulated proteins detected in *indPRPS1* samples, we found the regulator of fatty-acid composition 3 (RFC3; AT3G17170), which plays an important role in the plastid rRNA processing [77]. Many of these down-regulated proteins were located in chloroplast and nucleus (Fig 11B).

The significantly enriched GO terms, retrieved by analysing the up-regulated proteins detected in *indPRPL4* samples, were mainly associated with mitochondrial electron transport chain and mitochondrial ribosomes (S9 Table). Moreover, GO annotation revealed that several DAPs in *indPRPL4* seedlings belong to "response to stress" (GO:0006950), as also detected by the transcriptome analysis (Fig 11C). The GO annotation tool allowed to locate DAPs in *indPRPL4* plants especially in the nucleus, chloroplast and cytoplasm (Fig 11D). Overall, our results, in agreement with transcriptomic analysis, point out very specific proteomic changes following the alteration of the two ribosomal proteins PRPS1 and PRPL4. In fact, the induction of *PRPS1* leads to a general repression of translation, transcription and chloroplast organization. Conversely, the increased amount of PRPL4 protein promotes the cellular stress response, leading to a positive regulation of proteins involved in RNA metabolism and translation, with a major contribution also given by components of the mitochondrial metabolism.

Finally, to verify whether the differential gene expression overserved in *indPRPS1*+ DEX leaves was due to either increased *PRPS1* transcripts or decreased PRPS1 protein amount, we analysed the accumulation of *TULP5* and *SWEET13* (S14A Fig) transcripts (highly up-regulated in *indPRPS1* + DEX leaf discs) by comparing their expression in *indPRPS1* +DEX, in which the up-regulation of *PRPS1* transcripts is followed by PRPS1 protein degradation, to *indPRPS1 clpd-1* + DEX, in which *PRPSI* transcript upregulation is observed but PRPS1 protein degradation is inhibited (Figs 6 and S7). Interestingly, *TULP5* and *SWEET13* transcripts did not over-accumulate in the absence of a functional CLP protease, indicating that the re-orchestration of nuclear gene expression requires the accumulation/degradation of PRPS1 protein. Similarly, transgenic lines in which the *prps1-2* null allele is either under the control of *CaMV35S* promoter or the inducible promoter (see Materials and Methods) in Col-0 wild-type background did not show any virescent phenotype, supporting the notion that, in order to trigger its own degradation, the transient over-accumulation of PRPS1 protein is required (S14B Fig).

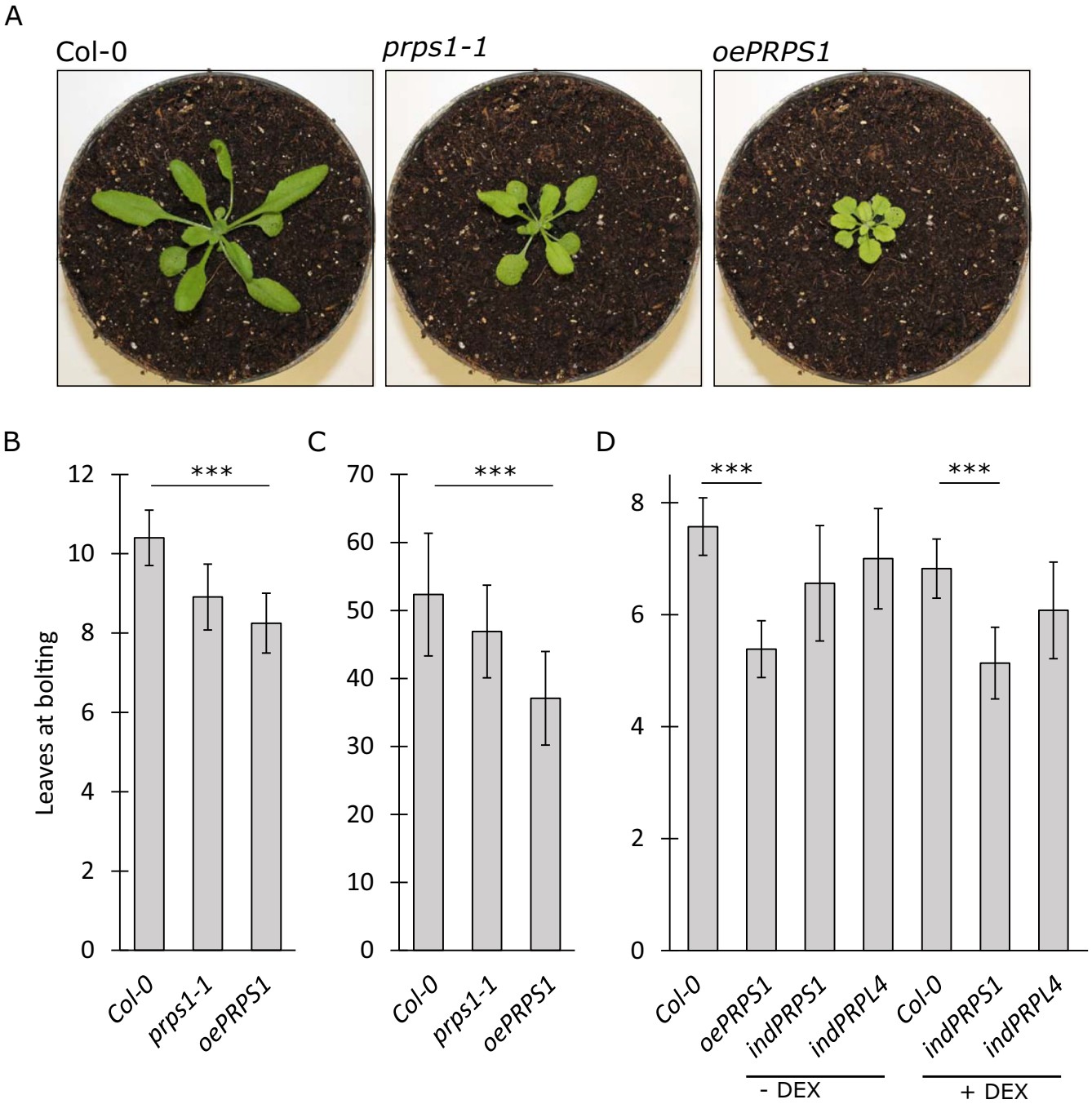

**Fig 9. Flowering time determination in Col-0, *prps1-1*, *oePRPS1*, *indPRPS1* (± DEX) and *indPRPL4* (± DEX) lines.** A) Pictures of representative plants at bolting grown on soil in long day conditions. B) Average numbers ± standard deviations of leaves at bolting observed in Col-0, *prps1-1* and *oePRPS1* plants grown on soil in long day conditions. C) Average numbers ± standard deviations of leaves at bolting observed in Col-0, *prps1-1* and *oePRPS1* plants grown on soil in short day conditions. D) Average numbers ± standard deviations of leaves at bolting observed in Col-0, *oePRSP1*, *indPRPS1* and *indPRPL4* plants grown on MS medium devoid (- DEX) or supplemented (+ DEX) with 2 μM dexamethasone in long day conditions. Statistical significance was calculated via Student's t-test (*** indicates $P < 0.001$).

A

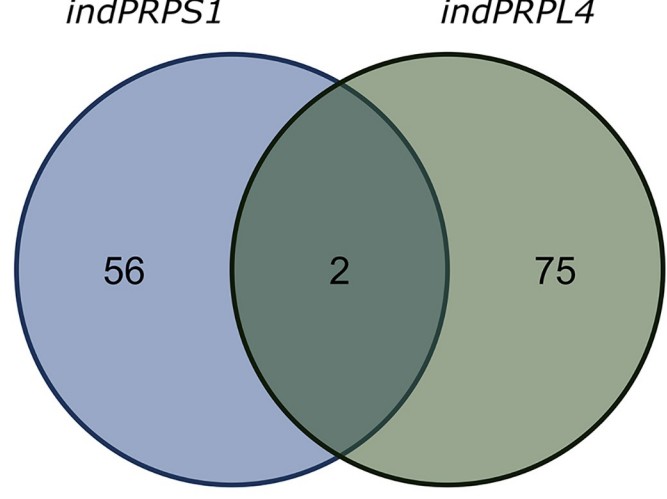

B

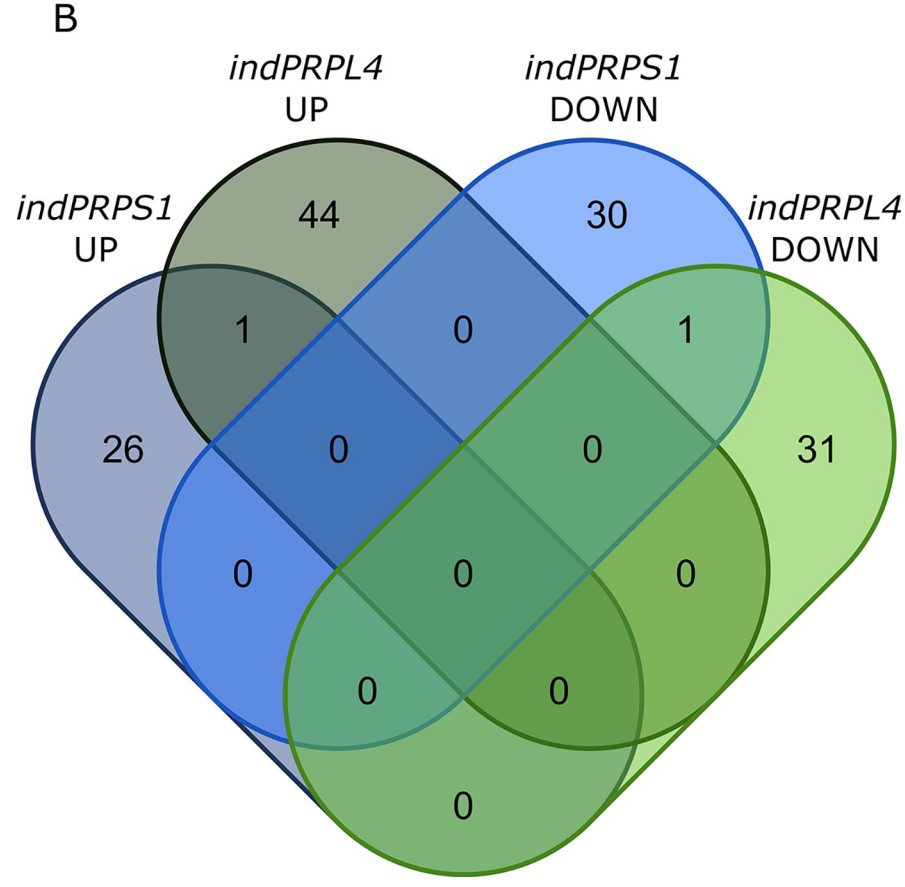

**Fig 10. Comparison of proteome from *indPRPS1* and *indPRPL4* lines infiltrated or not with DEX.** A) Venn diagram showing the number of DAPs found in *indPRPS1* and *indPRPL4* upon induction. B) Venn diagram showing the distribution of DAPs resulted from the comparison of *indPRPS1*UP, *indPRPL4*UP, *indPRPS1*DOWN and *indPRPL4*DOWN lists.

## Discussion

### Altered *PRPS1* expression impairs chloroplast biogenesis

*PRPS1* is a nuclear gene encoding the S1 protein of the plastid 30S small ribosomal subunits. Reduced expression of *PRPS1* gene results in diminished plastid translation and decreased photosynthetic efficiency [27,31], while its disruption arrests embryo development at the globular stage in Arabidopsis (see Fig 1) and prevents greening of seedlings in rice (*albino seedling lethality 4*; [78]). Clearly, these observations indicate that *PRPS1* is an essential gene required during early stages of chloroplast biogenesis, as reported in the case of other essential nuclear genes coding for plastid ribosomal proteins [27,35,79]. The apparent discrepancy between the early block of embryo development in Arabidopsis and the arrest of seedling greening in rice is in agreement with previous findings [80–83] and compatible with the essential nature of fatty acid biosynthesis during chloroplast biogenesis. In particular, the lack of the plastid-encoded accD-subunit of the multimeric acetyl-CoA carboxylase required for fatty acid biosynthesis is responsible for the lethality of Arabidopsis embryos defective in plastid translation [35]. In contrast, grass species contain a plastid-located monomeric acetyl-CoA carboxylase that, differently from Arabidopsis, is encoded in the nucleus and translated in the cytosol [84,85]. Therefore, fatty acid biosynthesis (and embryogenesis) can continue even when plastid protein synthesis is affected in these species.

Intriguingly, the reduced accumulation of S1 protein can also be obtained when *PRPS1* gene expression, and the consequent transcript accumulation, is both constitutively increased in *oePRPS1* and *indPRPS1* + DEX seedlings (Fig 2) and induced in *indPRPS1* leaf discs for 24 hours (Fig 4). This finding, together with the very limited increase in transcript accumulation observed in *oePRPS1* and *indPRPS1* + DEX seedlings, indicates the existence of post-transcriptional regulatory mechanisms aimed to prevent PRPS1 over-accumulation, as shown previously [31]. The strict control on *PRPS1* gene expression and protein accumulation appears to be rather specific and certainly it does not apply to *PRPL4*, a nuclear gene encoding a core subunit of the 50S plastid ribosome, also essential for embryo development and plant viability [27,35]. In fact, *oePRPL4* and *indPRPL4* + DEX lines were able to accumulate 25–30 times more transcripts, together with the double amount of PRPL4 protein, in comparison to Col-0 and *indPRPL4* − DEX controls (Figs 2 and 4).

Furthermore, the constitutive over-expression of *PRPS1* gene, driven in 16 DAS *oePRPS1* and *indPRPS1* + DEX plantlets, resulted in the impairment of chloroplast differentiation and physiology, leading to a virescent phenotype visible in the emerging leaves and in the younger portions of the leaves, corresponding to the tissue proximal to the petiole, but not in cotyledons, supporting the existence of major differences in the molecular mechanisms at the basis of plastid biogenesis, plastid protein homeostasis and degradation between cotyledons and leaves [51,52,86,87]. In particular, transmission electron microscopy (TEM) observations of the chlorotic tissues revealed the presence of cells displaying misshapen chloroplasts with altered thylakoid membrane ultrastructure and large plastoglobuli in the stroma (Fig 3), reported to be associated with the degradation of chlorophylls and thylakoids in response to abiotic and biotic stresses or during senescence [88–90]. Moreover, the presence of vesicles budding from chloroplasts and containing electron dense material, together with the observation of entire round chloroplasts inside the vacuole (Fig 3), indicates the ongoing chloroplast

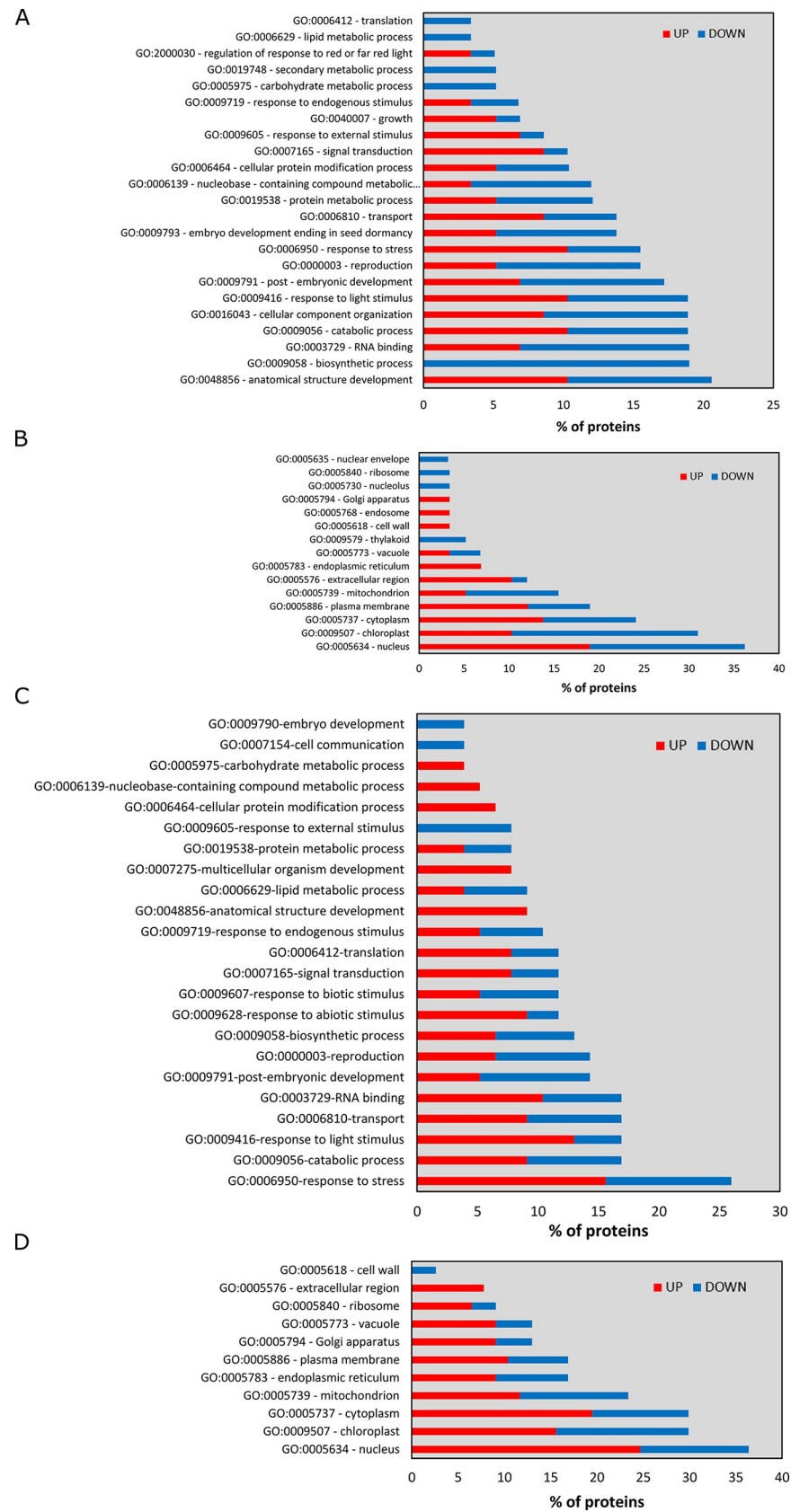

**Fig 11. Protein categorization based on GO terms.** Protein categorization based on Biological process GO terms of DAPs found in *indPRPS1* (A) and *indPRPL4* (C) samples and based on Cellular Content GO terms found in *indPRPS1* (B) and *indPRPL4* (D) samples. Blue and red bars correspond to proteins that were up or down accumulated, respectively.

degradation, similarly to what reported in literature [51,52,64,65]. Coherently, the relative expression of *ATI1* and *ATG8f* genes, involved in ATG-dependent micro-autophagy, was enhanced upon PRPS1 attempted over-accumulation (Figs 3 and S2A), whereas the expression of genes involved in ATG-independent micro-autophagy was mildly stimulated in *oePRPS1* (S2B Fig) [68]. Overall, these observations indicate that the reduced accumulation PRPS1 severely jeopardizes chloroplast integrity during leaf development, leading to the dismantling of damaged and misshapen chloroplasts with the final aim to remove reactive oxygen species–producing chloroplasts and redistributing nutrients to other tissues [34].

## A fully functional proteostasis machinery is needed to control PRPS1 accumulation in chloroplast stroma

All our attempts to increase the abundance of PRPS1 protein failed (Figs 2 and 4; [28]) and, together with that, the accumulation of other plastid ribosomal proteins, such as PRPL4 and PRPS5, was reduced, as observed in *indPRPS1* + DEX leaf discs (Fig 4), indicating that the decreased abundance of PRPS1 protein has a deleterious effect on plastid ribosome stability. In agreement with that, the pulse-labelling experiment conducted on *indPRPS1* leaf material infiltrated with DEX for 6 hours showed a severe inhibition of plastid protein translation and a larger fraction of PRPS1 protein freely associated to mRNA and not bound to actively translating ribosomes (Fig 4). These findings explain the defects in leaf greening observed in the different Arabidopsis lines (Figs 1 and 2) and support the role of PRPS1 as a stringently regulated translation factor rather than a "real" ribosomal protein, given its weakly and reversible association with the 30S subunit, similarly to previous observations in *E. coli* [16].

S1 protein is the closest PRPS1 homologue in *E. coli*, and it is encoded by the essential gene *rpsA* [11]. To gain possible insights on PRPS1 function, we attempted to rescue *E. coli* cell lethality due to S1 depletion and to phenocopy the bacteriostatic effects of S1 overaccumulation, by modulating the level of PRPS1 protein in *E. coli* cells (Fig 5). Our data clearly showed that PRPS1 is not able to functionally replace the endogenous S1, as cells depleted of *rpsA* but with moderate amount of PRPS1, failed to grow (Fig 5). According to previous studies, S1 protein exerts its functions in relationship with the specializations of its six S1 domains [19]. The interactions with the ribosome relies on S1 domains 1 and 2 [20], whereas the ability to bind mRNAs has been associated with the S1 domains 3, 4 and 5 [14]. As for domain 6, if removed together with S1 domain 5, the initiation of translation is hampered [19,25] and together with domain 4, it is implicated in ribosome dimerization and hibernation in stationary phase [26]. According to our *in silico* analysis, S1 domains 1 and 2 of PRPS1 appeared to be more similar to the respective domains 1 and 2 of S1, while domain 3 clustered together with domains 3, 4 and 5 of S1 (S4 Fig). The fact that PRPS1 is a smaller protein and possesses three out of the six S1 domains identified in S1, of which none could be associated with those required for translation initiation in *E. coli*, could explain the inability of PRPS1 to functionally replace S1 protein. Nonetheless, PRPS1 overaccumulation blocked cells growth as much as S1 (Figs 5 and S5). It has been shown that S1 over accumulation inhibits translation since the excess of "free" S1 interacts with mRNAs, preventing the ribosome loading [16,17,25,91]. However, PRPS1 was found both in ribosome-bound and -unbound fractions, suggesting that, although PRPS1 can interact with the ribosome core in *E. coli* cells, this interaction is rather unfruitful. In this

scenario PRPS1 would be able to inhibit *E. coli* growth by competing with the endogenous S1 protein and inhibiting ribosome activity (S5 Fig). Nevertheless, PRPS1 is likely capable of binding *E. coli* mRNAs with the S1 domains 2 and 3, since *E. coli* and plastid mRNAs share similar 5' UTRs with AU-rich sequences [92], making *E. coli* transcripts inaccessible to ribosomes and acting as a negative modulator of translation initiation. The inhibitory role of PRPS1 protein on *E. coli* protein synthesis is corroborated further by the fact that the accumulation of PRPS1 upon overexpression reaches very rapidly the plateau level and concomitantly leads to the decreased accumulation of S1 protein over time (Fig 5C), mimicking the role of S1 as feedback effector of its own regulation at the translational level [17].

Such regulatory mechanism has a different spatial constraint in photosynthetic eukaryotes due to the physical separation of the nuclear/cytosolic compartments, where the *PRPS1* transcripts and the precursor protein are synthesized, and the chloroplast stroma where the mature PRPS1 protein plays its functions. While the limited accumulation of *PRPS1* transcripts observed upon constitutive expression allows us to hypothesize the existence of a cytosolic post-transcriptional regulatory mechanism [32], our data strongly support the activation of a chloroplast stroma post-translational regulatory mechanism, in response to *PRPS1* overexpression, mediated by the plastid proteostasis machinery (Fig 6). In chloroplasts, the major soluble stromal protease is the CLP complex, composed of multiple nuclear-encoded subunits with the addition of ClpP1 subunit, the only one to be plastid-encoded [76]. Consequently, ClpP1 abundance is susceptible to genetic defects in plastid gene expression or to drugs inhibiting plastid translation [39]. The co-infiltration of DEX with the plastid translational inhibitor lincomycin allowed a larger accumulation of PRPS1 protein (Fig 6A). Accordingly, an increased PRPS1 protein accumulation (Fig 6C) could be achieved by introgressing the *prps1-1* knockdown allele into *prpl11-1* genetic background, in which the chloroplast translation is reduced [42]. This is also in line with the restoration of PRPS1 protein accumulation observed in *prps1 gun1* and *prps1 rh50* double mutants, as GUN1 stimulates the activity of the Nuclear-Encoded Polymerase (NEP) that, among other plastid house-keeping genes, is responsible of the transcription of *clpP1*, while RH50 is involved in plastid ribosome assembly and plastid translation [28,52,93–95]. Intriguingly, GUN1 was also found to physically interact with the plastid protein homeostasis machinery, including CLPC subunits [28]. Furthermore, a comparable increase in PRPS1 levels were also observed in the double mutants *prps1-1 clpc1-1* and *prps1-1 clpd-1* (Fig 6C), in *indPRPS1 clpc1-1* and *indPRPS1 clpd-1* (Fig 6D) and in *oePRPS1 clpc1-1* and *oePRPS1 clpd-1* (S7C Fig), in which the two plastid chaperones required by CLP protease to interact with the substrates are missing [44]. These pieces of evidence strongly point towards CLP protease complex as one of the main regulators of PRPS1 protein abundance in Arabidopsis chloroplasts.

Our findings are also in agreement with previous reports. In particular, by combining transcriptomic and proteomic analyses, Wu et al. were able to show that plastid ribosomal proteins are regulated post-translationally, suggesting a protein degradation-based mechanism [96]. Moreover, in *Chlamydomonas*, *CreS1* expression is induced by light, while CreS1 protein levels remain constant, pointing also in this case to the post-translational regulation of CreS1 abundance [50].

On the other hand, the constitutive over-accumulation of PRPL4 protein upon induction didn't affect neither the plastid translation nor the chloroplast ultrastructure (Figs 2–4). Interestingly, chaperones both resident in the cytoplasm and in plastids were progressively up-regulated as PRPL4 levels increased, indicating the activation of protein homeostasis mechanisms to cope with the increased protein amount (S11 Fig). These different responses to the overexpression of two plastid ribosomal proteins could be due to their intrinsic features and activities, having PRPS1 mRNA-binding properties which, if not kept under strict control, would have dramatic consequences on chloroplast functionality.

## *PRPS1* overexpression promotes early flowering while PRPL4 overaccumulation triggers cpUPR

The switch from vegetative growth to reproductive growth is a pivotal event in plant development, mostly dependent on the environmental stimuli such as day length (photoperiodic flowering) and temperature (vernalization) that plants perceive to determine the proper timing to assure successful reproduction, and ultimately the survival of the species [97]. However, plants subjected to a variety of stressful conditions can anticipate their flowering through a new category of flowering response, known as stress-induced flowering, aimed to guarantee species survival when they cannot adapt to unfavourable conditions [98]. Although the mechanistic details behind the stress-induced flowering are still not fully understood, hormones seem to be at least partially involved in these pathways as they are produced under stress and regulate gene expression to cope with it [98,99]. This seems to be the case of plants subjected to prolonged *PRPS1* overexpression and characterised by an early flowering phenotype regardless of day length (Fig 9). In these plants, the stress seems to be caused by the reduced chloroplast protein synthesis and the consequent accumulation of damaged proteins and, possibly, by low sucrose concentration due to the reduced photosynthetic performance (see Fig 2; [100]). Consistently, the transcriptomic profile of Arabidopsis lines characterised by the short-term induction of *PRPS1* gene highlighted that the early nuclear gene expression response to PRPS1 imbalance in chloroplasts is characterised by a robust enrichment of up-regulated genes associated with the reproductive phase transition and the cellular response to hormones, such as "photoperiodism, flowering", "vegetative to reproductive phase transition of meristem" and "cellular response to hormone stimulus" (Fig 8 and S3 Table), and by the concomitant repression, amongst others, of genes involved in "cell redox homeostasis", "response to wounding", "response to oxidative stress", "response to water deprivation". Accordingly, the circadian clock regulators *RVE8* and *CO*, the *IDD8* transcription factor and the Gibberellic acid enzyme *ga3ox1*, all positive modulators of flowering, are among the up-regulated genes. On the other hand, the tetratricopeptide thioredoxin-like protein *TTL4*, required for osmotic stress tolerance, *BASS5* and *BCAT4*, involved in glucosinolate biosynthesis, *PTR3*, required for defences against pathogens, and the transcription factor *NAC019*, involved in the response to dehydration, were found to be down-regulated upon *PRPS1* induction [101–105]. Interestingly, the mass spectrometry analysis of the tissue overexpressing *PRPS1* detected the increased accumulation of far-red insensitive 219 (FIN219; AT2G46370) and the phytochrome associated protein phosphatase 2C (PAPP2C; AT1G22280), both directly involved in the response to red, or far red light and in flower development [106,107]; while chloroplast-located proteins involved in "RNA metabolism" and "translation" where down-regulated.

In contrast, the results obtained from the transcriptomic and proteomic analyses performed on *indPRPL4* + DEX leaf discs were markedly different (Figs 7,8,10 and 11). The overaccumulation of PRPL4 promoted abiotic stress responses and protein homeostasis such as "response to hydrogen peroxide", "response to high light intensity", "response to heat", "response to oxidative stress" and "protein folding" (Fig 8C). In particular, transcription factors involved in stress responses such as *DREB2A*, *DREB2C* and *WRKY26* [108–110] were found to be upregulated, together with proteins directly involved in proteostasis maintenance, such as small HSPs, CLPB1 cytosolic unfoldase and the co-chaperone HOP3 [111–114]. Interestingly, *GOLS1* gene coding for a key enzyme of raffinose family sugar synthesis, shown to enhance plant resistance to oxidative damage, was also up-regulated [115,116]. These findings together with the increased accumulation of CLPB3, HSP90-1 and HSC70-4 proteins over time (see S11 Fig), corroborate the hypothesis that PRPL4 over-accumulation contributes to generate pressure on the plastid folding machinery that, in turns, activates retrograde signalling pathways

aimed at triggering a chloroplast-related UPR. It can be envisaged that the activated cpUPR contributes to the limited increase of PRPL4 protein abundance in Arabidopsis chloroplasts, i.e., two folds higher upon 24 HAI with respect to 0 HAI, despite the 30-fold increase of PRPL4 transcripts (Fig 4A and 4C).

Taken together these observations highlight the activation of different nuclear and cellular responses upon up-regulation of *PRPS1* and *PRPL4* gene expression and the consequent alteration of plastid protein homeostasis. Whereas the *PRPS1*-related response mainly relies on chloroplast breakdown with degradation and loss of chloroplast proteins, nucleic acids, pigments, lipids, and polysaccharides, aimed to promote the remobilization of resources, such as carbon and nitrogen, in favour of the anticipated reproductive phase, i.e. an escape strategy from a stress condition, the *PRPL4*-related response consists in the activation of a stress-adaptive pathway, compatible with the chloroplast UPR.

## Supporting information

**S1 Fig. Additional independent over-expressor and inducible lines carrying altered amount of PRPS1 and PRPL4 transcripts and proteins.** A) Visible phenotypes of 16 Days After Sowing (DAS) plantlets grown on MS medium devoid (- DEX) or supplemented (+ DEX) with 2 μM dexamethasone and their photosynthetic efficiency (*Fv/Fm*). Average values ± standard deviations. Scale bar = 3 mm. For simplicity reasons, the visible phenotypes of *oePRPL4* and *indPRPL4* lines, identical to Col-0 control plants are not shown. B) Immuno-blots of total protein extracts from Col-0, *prps1-1*, *oePRSP1*, *indPRPS1*, *oePRPL4* and *indPRPL4* 16 DAS plants grown on MS medium devoid (- DEX) or supplemented (+ DEX) with 2 μM dexamethasone. PRPS1 and PRPL4 specific antibodies were used for immuno-decoration. Coomassie Brilliant Blue (C.B.B.) stained gels are shown as loading control. Numbers show the relative protein abundance with respect to Col-0 (indicated as 100). Standard deviation was below ± 15%. One filter out of three biological replicates is shown. C) Relative expression values of *PRPS1* and *PRPL4* genes determined by qRT-PCR analyses of total RNA extracted from Col-0, *prps1-1*, *oePRSP1*, *indPRPS1*, *oePRPL4* and *indPRPL4* 16 DAS plants grown on MS medium devoid (- DEX) or supplemented (+ DEX) with 2 μM dexamethasone. Results of one out of three biological replicates are shown. Error bars indicate standard deviations of three technical replicates.
(EPS)

**S2 Fig. Expression analyses of genes involved in chloroplast micro-autophagy pathways.** A) Relative expression values of *ATG8f* and *ATI1* genes determined by qRT-PCR analyses of total RNA extracted from Col-0, *prps1-1*, *oePRSP1*, *indPRPS1* and *oePRPL4* 16 DAS plantlets grown on MS medium devoid (- DEX) or supplemented (+ DEX) with 2 μM dexamethasone. B) Relative expression values of *NBR1*, *NPC1*, *VAM3*, *VPS15*, *VPS25* and *CV* genes determined by qRT-PCR analyses of total RNA extracted from Col-0, *prps1-1*, *oePRPS1* and *oePRPL4* 16 DAS plantlets. Data from one out of three biological replicates are shown. Error bars indicate standard deviations of three technical replicates. Asterisks indicate statistical significance with respect to Col-0 as evaluated by Student's t-test (*$P < 0.05$; **$P < 0.01$; n.s.: not significant).
(EPS)

**S3 Fig. Isolation of PRPS1-containing complexes in leaf samples with different amounts of PRPS1 and PRPL4 proteins.** A) 16 DAS Col-0, *prps1-1*, *oePRPS1* and oe*PRPL4* leaf material was subjected to sucrose gradient fractionation aimed to isolate PRPS1-contaning complexes. Filters were then immuno-decorated with PRPS1 antibody to show the accumulation of PRPS1 in Low Molecular Weight (LMW) and High Molecular Weight (HMW) fractions. C.B.

B.-stained filters are shown as loading control. B) Quantification of PRPS1 accumulation in LMW (1–5) and HMW (6–9) fractions as evaluated by Image Lab software on representative blots. Each lane has been quantified as absolute value (see also Materials and Methods). The sum of all signals has been set as 1. The values reported in the graph are relative to the total sum for each condition (DEX -/+).
(EPS)

**S4 Fig. Phylogenetic tree generated from the multiple sequence alignment of S1 domains from S1 (Rps1, E. coli) and PRPS1 (A. thaliana) proteins.**
(EPS)

**S5 Fig. E. coli strains over accumulating S1 and PRPS1 proteins.** A) $OD_{600}$ measurements of *E. coli indrpsA* (white symbols; C-5691 strain) and *indPRPS1* (blue symbols; C-1a strain carrying *PRPS1-pQE31-pREP4* plasmid) strains grown under non-inducing (squares) or inducing (circles) conditions. Strains were sampled at 0, 30, 60, 90, 120, 150 and 180 Minutes After Induction (MAI). Data from one out of three biological replicates are reported. B) Detection of S1 or PRPS1 proteins in ribosome-unbound (SN) and ribosome-bound (P) fractions through immunoblotting from *indrpsA* or *indPRPS1 E. coli* extracts grown in absence or presence of 1 mM IPTG. L4 protein was probed as marker of ribosome-bound fraction. C.B.B.-stained gels are shown as loading control.
(EPS)

**S6 Fig. Growth rate measurements of Col-0, prps1-1, prpl11-1, clpc1-1, clpd-1, prps1-1 prpl11-1, prps1-1 clpc1-1 and prps1-1 clpd-1 genotypes grown under long-day conditions in a growth chamber for 21 days.** Leaf area is expressed as $mm^2$ (DAS, Days after sowing). Error bars indicate standard deviations of at least ten measurements.
(EPS)

**S7 Fig. PRPS1 protein accumulation and the stromal CLP protease.** A) Immunoblots of total protein extracts from *indPRPS1#2*, *indPRPS1#2 clpc1-1* and *indPRPS1#2 clpd-1* leaf discs sampled at 0, 3, 6 and 24 HAI with 2 μM dexamethasone. Filters were incubated with antibodies raised against PRPS1 protein. C.B.B.-stained gels are shown as loading control. Numbers below the immunoblots show the relative protein abundance with respect to 0 HAI (indicated as 100). Data are derived from one out of three biological replicates. Standard deviation was below 15%. B) Visible phenotypes of Col-0 and *prps1-1*, *oePRPS1*, *clpc1-1*, *oePRPS1 clpc1-1*, *clpd-1* and *oePRPS1 clpd-1* 20 DAS plants grown on soil. Scale bar = 1 cm. The photosynthetic efficiency of each genotype is shown as average values ± standard deviations. Statistical significance with respect to *oePRPS1* value was calculated via Student's t-test (* indicates $P < 0.05$). C) Immunoblot of total protein extracts from Col-0 and *prps1-1*, *oePRPS1*, *clpc1-1*, *oePRPS1 clpc1-1*, *clpd-1* and *oePRPS1 clpd-1* mutants obtained by using a PRPS1 specific antibody. The immunoblot incubated with Histone H3 specific antibody and a C.B.B.-stained gel are shown as loading control. Numbers below the PRPS1 immunoblot show the relative protein abundance with respect to Col-0. Standard deviation was below 15%.
(EPS)

**S8 Fig. Subcellular localization of PRPS1 protein.** Immunoblots of proteins from the plastid-enriched fraction and the soluble extra-plastidial fraction isolated from Col-0, *prps1-1* and *oePRPS1* 16 DAS plantlets grown on soil. Filters were incubated with PRPS1-specific antibody. Specific antibodies raised against LHCB4 and HSP90-1 proteins were employed as markers for the plastid-enriched fraction and the soluble fraction, respectively.
(EPS)

**S9 Fig. Visible phenotype and photosynthetic efficiency of leaf discs harvested from indPRPS1 and indPRPL4 lines, vacuum-infiltrated and incubated for 6 hours either in the absence or presence of DEX.** A) Representative pictures of leaf discs after the treatments. B) Measurements of the photosynthetic efficiency parameter *Fv/Fm* after the treatment. Overall, no significant changes were observed.
(EPS)

**S10 Fig. Transcriptome PCA analysis of indPRPS1 (upper plot) and indPRPL4 (lower plot) samples in + DEX (green) or − DEX (orange) conditions.**
(EPS)

**S11 Fig. Immunoblots of total protein extracts from indPRPL4 and indPRPS1 leaf discs.** A) Proteins from *indPRPL4* leaf discs sampled at 0, 3, 6 and 24 HAI with 2 μM dexamethasone were subjected to immunoblot analyses using antibodies specific for CLPB3 plastid unfoldase, HSP90-1 and HSC70-4 cytosolic chaperones. B) Immunoblots of total protein extracts from *indPRPS1* leaf discs, sampled and analysed as in A. C.B.B.-stained gels are shown as loading control. Numbers below the immunoblots show the relative protein abundance with respect to 0 HAI (indicated as 100). Data from one out of three biological replicates are shown. Standard deviation was below 15%.
(EPS)

**S12 Fig. Expression analyses of PEP- and NEP-dependent plastid-encoded genes.** Relative expression values of *accD*, *rpoA*, *rpoB*, *rps12*, *clpP1*, *tic214*, *psaA*, *psbA* and *rbcL* genes determined by qRT-PCR analyses of total RNA extracted from leaf discs harvested from *indPRPS1* and *indPRPL4* lines and incubated for 6 hours either in the absence or presence of DEX after vacuum-infiltration. Error bars indicate standard deviations of three biological replicates. No significant differences were detected.
(EPS)

**S13 Fig. Proteome PCA analysis of indPRPS1 (upper plot) and indPRPL4 (lower plot) samples in + DEX (green) or − DEX (orange) conditions.**
(EPS)

**S14 Fig. Accumulation of PRPS1 transcript alone is not sufficient to trigger the visible and molecular phenotypes observed in indPRPS1 plantlets.** A) Relative expression values of *PRPS1*, *TULP5* and *SWEET13* genes determined by qRT-PCR analyses on total RNA extracted from leaf discs of *indPRPS1* and *indPRPS1 clpd-1* lines, vacuum-infiltrated and incubated for 6 hours either in the absence or presence of DEX. Results of one out of three biological replicates are shown. Error bars indicate standard deviations of three technical replicates. Statistical significance with respect to–DEX sample was calculated via Student's t-test (* indicates $P < 0.05$; ** $P < 0.01$). B) Visible phenotypes of 16 DAS plantlets carrying either the wild-type *PRPS1* allele or the null *prps1-2* allele, grown on MS medium devoid (- DEX) or supplemented (+ DEX) with 2 μM dexamethasone. Scale bar = 3 mm.
(EPS)

**S1 Table. List of oligonucleotides used in this work.** Gene targets, features and sequences are indicated.
(XLSX)

**S2 Table. List of Differentially Expressed Genes (DEGs) for indPRPS1 +DEX vs -DEX and indPRPL4 +DEX vs –DEX.** For each DEG the associated AGI code, the logarithmic fold-change (logFC), the gene name and a brief description of gene function are indicated. Only

DEGs with FDR < 0,05 are listed. In addition, common and unique DEGs from the two lists, up- or down-regulated, are summarized. A) Unique up-regulated DEGs found in *indPRPS1*; B) Unique up-regulated DEGs found in *indPRPL4*; C) Common up-regulated DEGs found in *indPRPS1* and *indPRPL4*; D) Unique down-regulated DEGs found in *indPRPS1*; E) Unique down-regulated DEGs found in *indPRPL4*; F) Common down-regulated DEGs found in *indPRPS1* and *indPRPL4*; G) up-regulated DEGs found in *indPRPL4* but down-regulated in *indPRPS1*.
(XLSX)

**S3 Table. List of enriched Gene Ontology (GO) terms resulted from the analysis of unique up-regulated DEGs found in indPRPS1 +DEX vs -DEX comparison.** Each GO term listed is associated with a description, the number of genes found in the input list, the total amount of genes found in the reference, the p-value, the FDR and the fold-enrichment (FE). The list of the associated DEGs with a brief description is also reported for each GO term.
(XLSX)

**S4 Table. List of enriched GO terms resulted from the analysis of unique down-regulated DEGs found in indPRPS1 +DEX vs -DEX comparison.** Each GO term listed is associated with a description, the number of genes found in the input list, the total amount of genes found in the reference, the p-value, the FDR and the fold-enrichment (FE). The list of the associated DEGs with a brief description is also reported for each GO term.
(XLSX)

**S5 Table. List of enriched GO terms resulted from the analysis of unique up-regulated DEGs found in indPRPL4 +DEX vs -DEX comparison.** Each GO term listed is associated with a description, the number of genes found in the input list, the total amount of genes found in the reference, the p-value, the FDR and the fold-enrichment (FE). The list of the associated DEGs with a brief description is also reported for each GO term.
(XLSX)

**S6 Table. List of Differentially Accumulated Proteins (DAPs) for indPRPS1 +DEX vs -DEX and indPRPL4 +DEX vs -DEX.** For each DAP, the associated AGI code, the logarithmic fold-change (logFC), the gene name and a brief description of gene function are indicated.
(XLSX)

**S7 Table. List of enriched Biological Process GO terms resulted from the analysis of down-regulated DAPs found in indPRPS1 +DEX vs -DEX comparison.** Each GO term listed is associated with a description, the number of genes found in the input list, the total amount of genes found in the reference, the FDR and the fold-enrichment (FE).
(XLSX)

**S8 Table. List of enriched Cellular Component GO terms resulted from the analysis of down-regulated DAPs found in indPRPS1 +DEX vs -DEX comparison.** Each GO term listed is associated with a description, the number of genes found in the input list, the total amount of genes found in the reference, the FDR and the fold-enrichment (FE).
(XLSX)

**S9 Table. List of enriched Cellular Component GO terms resulted from the analysis of up-regulated DAPs found in indPRPL4 +DEX vs -DEX comparison.** Each GO term listed is associated with a description, the number of genes found in the input list, the total amount of genes found in the reference, the FDR and the fold-enrichment (FE).
(XLSX)

## Acknowledgments

We are grateful to James Friel for critical reading of the manuscript and English editing. We are thankful to Lucio Conti and Cecilia Zumajo for fruitful discussion on flowering time regulatory mechanisms. We are also grateful to Norma Lattuada, Roberto Ferrari, Valerio Paravicini and Mario Beretta for excellent technical assistance. NoLimits platform at University of Milano is acknowledged for TEM analyses.

## Author Contributions

**Conceptualization:** Luca Tadini, Nicolaj Jeran, Federica Briani, Candida Vannini, Paolo Pesaresi.

**Data curation:** Luca Tadini, Nicolaj Jeran, Guido Domingo, Federico Zambelli, Anna Calabritto.

**Formal analysis:** Luca Tadini.

**Investigation:** Luca Tadini, Guido Domingo, Simona Masiero, Federica Briani, Candida Vannini, Paolo Pesaresi.

**Methodology:** Luca Tadini, Guido Domingo, Federico Zambelli, Elena Costantini, Sara Forlani, Milena Marsoni.

**Project administration:** Simona Masiero.

**Software:** Federico Zambelli.

**Supervision:** Luca Tadini, Simona Masiero, Paolo Pesaresi.

**Validation:** Anna Calabritto.

**Writing – original draft:** Luca Tadini, Nicolaj Jeran, Guido Domingo.

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
