## [Decision Letter · Decision Letter 0]

26 Aug 2022

Dear Dr Pesaresi,

Thank you very much for submitting your Research Article entitled 'Perturbation of protein homeostasis brings plastids at the crossroad between repair and dismantling' to PLOS Genetics.

The manuscript was fully evaluated at the editorial level and by three independent peer reviewers. The reviewers appreciated the attention to an important problem, but raised some substantial concerns about the current manuscript. Based on the reviews, we will not be able to accept this version of the manuscript, but we would be willing to review a much-revised version. We cannot, of course, promise publication at that time.

If you decide to revise the manuscript for further consideration at PLOS Genetics, please aim to resubmit within the next 60 days, unless it will take extra time to address the concerns of the reviewers, in which case we would appreciate an expected resubmission date by email to plosgenetics@plos.org.

[LINK]

We are sorry that we cannot be more positive about your manuscript at this stage. Please do not hesitate to contact us if you have any concerns or questions.

Yours sincerely,

Li-Jia Qu

Section Editor

PLOS Genetics

Reviewer's Responses to Questions

**Comments to the Authors:**

Reviewer #1: This study tried to understand how the chloroplast proteome is maintained. To this end, the author investigated the role of two plastid ribosomal proteins PRPS1 and PRPL4 in protein homeostasis through deletion and overexpression of both proteins. Although the author found that the changes in the abundance of PRPS1 and PRPL4 may result in a perturbation of protein homeostasis using the analyses of transcriptomics, proteomics and transmission electron microscopy, this perturbation may not occur in vivo. In addition, the results in this study did not provide novel points as claimed by the author. I do not see any advanced understanding of molecular mechanisms about chloroplast retrograde communication and new insight into cellular responses to impaired plastid protein homeostasis as claimed by the author.

Reviewer #2: The authors aimed to address chloroplast quality-control pathway by modulating the expression of PRPS1 and PRPL4 and described protein homeostasis in chloroplast. I think the transcription level and protein level of chloroplast-encode genes should be paid attention to reflect the state of the chloroplast in the transcriptome and proteome, which is important to understand the state of protein homeostasis in chloroplast. However, no data on this point has been reported in the manuscript. Here I have some concerns which should be addressed.

1. The authors showed that the over-expression of PRPS1 plant show slightly pale true levels, but the authors use the indPRPS1 and indPRPL4 plants infiltrated in presence of DEX for 6 hours in the transcriptome experiment and proteome experiment. I don’t know the reason for this? The authors should show the phenotype of the indPRPS1 plant infiltrated in presence of DEX for 6 hours. In the proteome analysis, chloroplast-encode proteins were not changed. However, in the Fig4B, when the indPRPS1 plants are infiltrated in presence of DEX for 6 hours, the amount of PRPS1 protein was significantly reduced. At this point, was chloroplast not impaired? Or could the author explain this?

2. The transcript levels of chloroplast-encode genes were not altered in the transcriptome. The transcriptome analysis methods should be described in more detail. Especially, how to build the libraries. This is related to the quantification of chloroplast gene.

3. PRPS1 as a member of plastid ribosome, the amount of PRPS1 protein was significantly reduced when the indPRPS1 plants infiltrated in presence of DEX for 6 hours. I think the chloroplast may not be under the normal condition, the transcription level and translation level still remained unchanged? And the author also investigated the possible negative effect of PRPS1 inducible expression on plastid protein translation. In this paper, I think the transcription level and protein level of chloroplast-encode genes should be shown because the authors aimed to describe the effect of perturbance chloroplast protein homeostasis.

4. In the Fig4D, the contents of RbcL and D1/D2 showed a big difference in vivo. The blots may be over exposed in order to quantitative analysis. The author can check the two proteins in different blot.

5. The author think PRPS1 accumulation is negatively regulated by chloroplast Clp protease complex. The accumulation of PRPS1 protein increased about two-fold in all the double mutants tested with respect to prps1-1. In the Fig6C, the CBB stain is not precise in my eyes. Thought I believe the results, the authors should check the results of protein quantification by examining gene Actin.

6. As for the overexpression PRPS1 material, I am also curious the protein level of PRPS1 was dramatically decreased. This point should be clarified. One point is that if all PRPPS1 proteins can enter the chloroplast, or part of them stay in the cytoplasm and then be degraded. Or is it likely the transgenetic plants carry a second mutation?

Reviewer #3: In the manuscript “Perturbation of protein homeostasis brings plastids at the crossroad between repair and dismantling”, authors addressed a chloroplast quality-control pathway by modulating the expression of two nuclear genes encoding plastid ribosomal proteins PRPS1 and PRPL4. Plastid protein homeostasis is maintained through the balance between de novo synthesis and degradation and is regulated by both plastid- and nuclear-encoded proteins. The research topic is of great interest to the readers of PLOS Genetics. Thus, the subject is relevant and deserves to be published in an important journal such as PLOS Genetics upon clarification of the following points.

Major points:

1. The lack of genetic material makes the genetic data of this manuscript unconvincing. The transgenic plants (oePRSP1, indPRPS1, oePRPL4 and indPRPL4) used in this paper have only one line respectively. Please provide other independently transformed transgenic lines and their corresponding data, including the phenotypes and the expression levels of relevant RNA and protein.

2. The authors concluded that “the accumulation of PRPS1 is negatively regulated by chloroplast CLP protease complex” by comparing the accumulation of PRPS1 in the prps1-1 mutant and prps1-1 clp double mutants. prps1-1 is a knockdown mutant of PRPS1, and the increased amount of PRPS1 protein in prps1-1 clp double mutants is not sufficient to draw this conclusion. Why is the accumulation of PRPS1 protein not higher in clp mutants than in wild-type plants? This problem can be better explained if indPRPS1 is crossed with the clp mutants.

3. To investigate the molecular responses to the increased expression of PRPS1 gene, transcriptome and proteome analysis was performed on leaf discs harvested from indPRPS1 that were vacuum infiltrated for 6 hours in either the absence or presence of DEX. As shown in Figure 4, after 6 hours DEX induction, the RNA expression level of PRPS1 increased 50-fold, while the protein level of PRPS1 decreased to 40%. Therefore, it is difficult to tell whether the transcriptomic and proteomic results are caused by increased PRPS1 gene expression or decreased PRPS1 protein content. Figure 2 showed that compared to WT, the expression level of PRPS1 gene is 20% and the protein content is 45% in prps1-1 mutant. If the prps1-1 mutant is added as a reference in transcriptome and proteome analysis, it will be helpful to draw the correct conclusion.

Minor points:

1. In Figure 2, the alteration of PRPS1 expression has different effects on true leaves and cotyledons. Please explain it.

2. Please add information about prps1-1 mutants.

3. There are three pathways of chloroplast degradation: senescence-associated vacuoles (SAVs), chloroplast vesiculation, and autophagy. In order to accurately analyze chloroplast degradation in Figure 3, marker lines of different degradation pathways should be added as controls, or marker genes of different degradation pathways should be comprehensively examined.

4. Please explain the calculation method of Figure 4F.

**Have all data underlying the figures and results presented in the manuscript been provided?**

Reviewer #1: Yes

Reviewer #2: Yes

Reviewer #3: Yes

PLOS authors have the option to publish the peer review history of their article (what does this mean?). If published, this will include your full peer review and any attached files.

Reviewer #1: No

Reviewer #2: No

Reviewer #3: **Yes: **Xin Hou

---

## [Decision Letter · Decision Letter 1]

9 May 2023

Dear Dr Pesaresi,

We are pleased to inform you that your manuscript entitled "Perturbation of protein homeostasis brings plastids at the crossroad between repair and dismantling" has been editorially accepted for publication in PLOS Genetics. Congratulations!

Yours sincerely,

Li-Jia Qu

Section Editor

PLOS Genetics

Li-Jia Qu

Section Editor

PLOS Genetics

Comments from the reviewers (if applicable):

Reviewer's Responses to Questions

**Comments to the Authors:**

Reviewer #2: The authors have addressed my concerns, and thus I suggest it to be accepted for publication.

Reviewer #3: Authors have thoroughly addressed my comments. I have no further remarks.

**Have all data underlying the figures and results presented in the manuscript been provided?**

Reviewer #2: Yes

Reviewer #3: Yes

PLOS authors have the option to publish the peer review history of their article (what does this mean?). If published, this will include your full peer review and any attached files.

Reviewer #2: No

Reviewer #3: **Yes: **Xin Hou

**Data Deposition**

http://datadryad.org/submit?journalID=pgenetics&manu=PGENETICS-D-22-00837R1

**Press Queries**

---

## [Editor Report · Acceptance letter]

25 Jun 2023

PGENETICS-D-22-00837R1 

Perturbation of protein homeostasis brings plastids at the crossroad between repair and dismantling 

Dear Dr Pesaresi, 

We are pleased to inform you that your manuscript entitled "Perturbation of protein homeostasis brings plastids at the crossroad between repair and dismantling" has been formally accepted for publication in PLOS Genetics! Your manuscript is now with our production department and you will be notified of the publication date in due course.

With kind regards,

Olena Szabo

PLOS Genetics

On behalf of:
